# Giant dust particles at Nevado Illimani: a proxy of summertime deep convection over the Bolivian Altiplano

Filipe G. L. Lindau[1], Jefferson C. Simões[1,2], Barbara Delmonte[3], Patrick Ginot[4], Giovanni Baccolo[3], Chiara I. Paleari[3], Elena Di Stefano[3], Elena Korotkikh[2], Douglas S. Introne[2], Valter Maggi[3], Eduardo Garzanti[3], and Sergio Andò[3]

[1]Centro Polar e Climático, Universidade Federal do Rio Grande do Sul, Porto Alegre, 91501-970, Brazil
[2]Climate Change Institute, University of Maine, Orono, ME 04469, USA
[3]Environmental and Earth Sciences Department, University Milano-Bicocca, Milan, 20126, Italy
[4]Univ. Grenoble Alpes, CNRS, IRD, Grenoble INP, IGE, 38000 Grenoble, France

**Correspondence:** Filipe G. L. Lindau (filipelindau@hotmail.com)

**Abstract.** A deeper understanding of past atmospheric circulation variability in the Central Andes is a high-priority topic in paleoclimatology, mainly because of the necessity to validate climate models used to predict future precipitation trends and to develop mitigation and/or adaptation strategies for future climate change scenarios in this region. Within this context, we here investigate an 18-years firn core drilled at the Nevado Illimani in order to interpret its mineral dust record in relation to seasonal
processes, in particular atmospheric circulation and deep convection. The core was dated by annual layer counting based on seasonal oscillations of dust, calcium and stable isotopes. Geochemical and mineralogical data show that dust is regionally-sourced in winter and summer. During austral summer (wet season) an increase in the relative proportion of giant dust particles ($\varnothing > 20\,\mu\text{m}$) is observed, in association with oscillations of stable isotope records ($\delta$D, $\delta^{18}$O). It seems that at Nevado Illimani both the deposition of dust and the isotopic signature of precipitation are influenced by atmospheric deep convection, which is
also related to the total amount of precipitation in the area. This hypothesis is corroborated by regional meteorological data. The interpretation of giant particle and stable isotope records suggests that downdrafts due to convective activity promote turbulent conditions capable of suspending giant particles in the vicinity of the Nevado Illimani. Giant particles and stable isotopes, when considered together, can be therefore used as a new proxy for obtaining information about deep convective activity in the past.

## 1    Introduction

Climate variability in the Central Andes and the Bolivian Altiplano has a strong link with atmospheric circulation and rainfall anomalies over the rest of tropical South America (e.g., Vuille, 1999). Over the Altiplano, a semi-arid plateau in the Central Andes with a mean elevation of 3800 m above the sea level (a.s.l.) (Fig. 1), climate variations have a direct effect on the availability of water resources, with severe economic and social impacts (Garreaud and Aceituno, 2001). The recent retreat of
Andean glaciers due to global climate change (Rabatel et al., 2013) poses issues not only for water availability (Soruco et al., 2015), but also for the preservation of glaciers as natural archives that could be soon lost. For example, in the period between

years 1963 and 2009, Nevado Illimani (Fig. 1) lost approximately 35% (9.49km$^2$) of its total area (Ribeiro et al., 2013). On Quelccaya ice cap (13°54'S, 70°48'W, 5670 m a.s.l., Fig. 1), the seasonal variations of stable isotopes began to deteriorate because of the percolation of meltwater through firn, affecting the record corresponding to the latter half of the twentieth century,

although the seasonality of the dust record is still preserved (Thompson et al., 2017).

Precipitation on the Bolivian Altiplano is largely concentrated in the summer months (Garreaud et al., 2003), in response to the peak phase of the South American Summer Monsoon (SAMS). During summer (December–January–February, DJF), the intensification and southward displacement of the Bolivian Anticyclone (thereafter referred as Bolivian High) promotes strong easterly winds and a turbulent entrainment of easterly air masses over the Andean ridge. In addition, the upward motion over

30 western Amazon, which is part of the meridional circulation between the tropical North Atlantic and western tropical South America, leads to increased convection and reduced tropospheric stability over the Central Andes (Segura et al., 2020). Such an atmospheric context favours the establishment of an eastward upslope air-flow and the advection of moisture from the Amazon basin toward the Andes (Zhou and Lau, 1998). Accordingly, summer precipitation is strongly associated with deep convective activity, enhanced by high amounts of water vapor in the boundary layer which destabilize the tropospheric column over the

35 Altiplano (Garreaud, 1999). Precipitation during the rest of the year is scarce: during winter months (June–July–August, JJA) the Altiplano is generally very dry, and advection of dry air from the Pacific region is promoted, being the advection of moist air from the east suppressed. Strong winter westerly winds and dry conditions allow massive local transport and deposition of dust over the Central Andean glaciers, and for this reason high seasonal contrast exists between wet summer and dry winter snow layers in terms of dust and aerosol content (Knüsel et al., 2005).

Besides seasonal variability, year-to-year climate over the Altiplano is also influenced by conditions in the tropical Pacific Ocean. During warm phase of the El Niño-Southern Oscillation (ENSO), the Altiplano climate is dry. Dry summers associated with El Niño events in the tropical Pacific are characterized by enhanced westerly flow over the tropical Andes inhibiting moisture advection from the Amazon basin (Knüsel et al., 2005; Thompson et al., 2013). Conversely, wet summers associated with a cooling of the tropical Pacific (La Niña events) promote further ingression of humid easterly air masses from the Amazon

Basin.

Developing an annually-resolved ice core record from the Altiplano is an opportunity to enhance our knowledge about present and past climate variability in the tropical Andes region. Previous ice core studies from the Central Andes (Correia et al., 2003; Knüsel et al., 2005; Osmont et al., 2019) reveal that the aerosol content of ice is dominated by local (i.e. glacier basins from Nevado Illimani) and regional (the Altiplano's area) mineral dust during the winter, when black carbon from biomass burning

in the Amazon basin is also present. During the summer, conversely, the concentration of aerosol and particulate matter is much lower while impurities of anthropogenic origin (e.g. Cu, As and Cd) are observed in higher proportions (Correia et al., 2003).

With the aim to enhance our knowledge about past and present climate variability in the tropical Andes region, a new shallow firn core (23.8 m long) was drilled on the Nevado Illimani (Eastern Cordillera, Central Andes), as an integration of the Ice

Memory project (https://www.ice-memory.org). In this study we investigate mineral dust aerosol variability and provenance in this firn core, through the analysis of dust concentration, grain size, geochemistry and mineralogy. The very pronounced

seasonal variations of the analyzed proxies allowed for the development of a precise chronology, which covers the 1999–2017 period, and for the investigation of correlation between dust records and other proxies. Dust particles entrapped in firn samples seem to originate from regional sources during both winter and summer, despite minor mineralogical differences between the two seasons are observed.

An interesting result concerns the presence of giant dust particles (presenting a diameter larger than 20 μm), whose relative variability (compared to the smaller particles) is correlated to the stable isotope record. Very large mineral dust particles were generally neglected in climate studies and underrepresented or non-represented in global climate models, because of their generally local origin with respect to the sampling site, and their relatively low number concentration (Albani et al., 2014; Adebiyi and Kok, 2020). The recent observation of such large dust grains even at great distance from the source puts into question the physical models used to estimate settling velocities, and suggests some additional mechanisms such as strong turbulence and upper-level outflow are needed to keep these dust particles aloft (van der Does et al., 2018). As a consequence, there is now a growing interest in such relatively less abundant but volumetrically important dust grains, which can play an important role in biogeochemical cycles, in cloud microphysics, in the ocean carbon cycle and atmospheric radiation budget (van der Does et al., 2018; Ryder et al., 2019). A few studies have also considered large mineral particles in snow and ice, obtaining interesting results, in particular related to the relationships existing between coarse particles and the atmospheric patterns responsible for their deflation, transport and deposition (Kutuzov et al., 2016; Wu et al., 2009, 2010; Simonsen et al., 2019).

Our data show that the proportion of giant dust particles into firn is correlated with local meteorological observations, and in particular with atmospheric deep convection over the Bolivian Altiplano during summer. This study shows for the first time that climatic processes control the presence of giant dust particles in Andean firn and ice. We found clear evidence that the convective activity over the Altiplano, reconstructed through the analysis of giant particles, is enhanced during summer periods, in agreement with observations concerning atmospheric circulation anomalies in the area (Vuille, 1999). From this perspective, this study demonstrates the great potential of giant particles records, which are strongly influenced by climatic and meteorological processes at high altitude continental glaciers. This is a first exploratory work, analysis of a longer ice core would be desirable in the future to investigate the relationships between giant dust particles deposition, atmospheric deep convection and periodic climatic phenomena (La Niña).

## 2 Material and Method

### 2.1 Field campaign and firn core sampling

Nevado Illimani (16°37'S, 67°46'W, 6438 m a.s.l.) is located 50 km southeast of the Bolivian capital, La Paz, and 180 km southeast of Lake Titicaca (Fig. 1). Its approximate dimensions are 10 per 4 km, with some peaks above 6,000 m a.s.l. Nevado Illimani consists of a granodiorite pluton of Late Oligocene age, with a short belt composed by a coeval dacitic flow located near the south-western border of the pluton (McBride et al., 1983; Jiménez and López-Velásquez, 2008). In June 2017, a 23.8 m firn core (corresponding to 13.75 m water equivalent) was drilled at an altitude of 6350 m a.s.l. on the saddle between the

two Nevado Illimani summits, approximately where two deep ice cores were recovered in June 1999 (Knüsel et al., 2003). The expedition was coordinated by a French, Russian, Bolivian and Brazilian team and integrated the Ice Memory project (Université Grenoble Alpes Foundation). An EM-100-1000 electromechanical ice core drill (Cryosphere Research Solutions, Columbus, Ohio, USA) was used for the drilling and three cores were extracted, two down the bedrock (136 and 134 m) and the core for this study (23.8 m).

The core (diameter of 10 cm), consisting of 24 sections of approximately 1 m length, was transported by mountain porters from the drilling site to the base camp during the night in order to prevent melting. Once at the base camp, the core sections were immediately transported to a refrigerated container located in La Paz, where temperature was set at -20 °C. After the drilling campaign, the container was shipped to the Institut des Géosciences de l'Environnement (IGE, Université Grenoble Alpes, France) where the core sections were weighted and cut longitudinally using a vertical band saw in a cold room (at -15 °C).

The stratigraphy of the firn core shows ∼1 to ∼5 cm thick layers with greater density (visually detected) distributed through-out it. They may be ice layers and/or wind crusts, and the cause of each layer is difficult to investigate by visual stratigraphy alone (Kinnard et al., 2008; Inoue et al., 2017). Thus, these features probably indicate events such as meltwater percolation, potentially affecting the core record by post-depositional changes. Ice/crust layers were counted and logged, being present in 37% of the 464 samples produced for dust analysis. Fig. S1 shows the distribution of the ice/crust layers observed in the firn core, along with the records of the giant particles percentage in terms of number (GPPnb, defined in Sect. 2.2) and $\delta$D. These layers show no clear correspondence with the depth intervals showing peak values and/or reduced seasonality in both records. Depth intervals with multiple ice/crust layers show a similar variability for both GPPnb and $\delta$D when compared with intervals showing few of these layers (Fig. S1). Thus, we consider that post-depositional processes related to the formation of ice/crust layers had little influence on the proxies registered in the firn core.

One quarter of the original core was dedicated to dust analyses and transported for this purpose to the EUROCOLD facility of the University of Milano-Bicocca (Italy). There, firn sections were transversely cut at 5 cm using a horizontal band saw with a cobalt steel blade and 464 samples were obtained. These were manually decontaminated by mechanical scraping with a clean ceramic knife inside a laminar flow high-efficiency particle air (HEPA) ISO 5 Class bench located in an ISO 6 Class cold room. Once decontaminated, the samples were put into clean Corning® centrifuge tubes and kept frozen until the measurements.

## 2.2 Coulter counter analysis

Samples were melted at room temperature, and a ∼10 mL aliquot from each was transferred to an Accuvette Beckman Coulter vial, previously washed with Millipore Q-POD® Element ultra-pure water (in an ISO 5 Class laminar flow bench located inside an ISO 6 Class clean laboratory). Each sample was treated following standardized protocols (Delmonte et al., 2002). A Beckman Coulter Multisizer 4 equipped with a 100 μm orifice was used to measure dust concentration and grain size (400 size channels within the 2–60 μm interval of spherical equivalent diameter). Samples were continuously stirred until the moment of the analysis, as the larger particles tend to settle rapidly. Systematic analysis of ultra-pure water blanks allows estimating a mean signal to noise ratio around 97. Each sample was measured twice, consuming 0.5 mL per measurement. The mean

relative standard deviation (RSD) between these two measurements considering both the number and the mass of particles was 7% and 29%, respectively.

The higher deviation for the mass in comparison to the total number of particles was expected due to the presence of heavy giant particles having diameters $>20\ \mu$m (coarse silt), for which small differences in size estimation lead to higher uncertainties. Indeed, when considering only the giant particles the mean RSDs were 55% and 63% for the number and mass distributions, respectively. Thus, the proportion (%) of giant particles (GPPnb) as well as total particle concentration, were calculated from the number size distribution. Approximately 14% of the samples showed very large uncertainties (RSD $>$ 100%) for GPPnb and were discarded. The mean RSD for GPPnb was 45%.

## 2.3  Instrumental Neutron Activation Analysis

A set of 10 samples was dedicated to Instrumental Neutron Activation Analysis (INAA). Samples were selected from different depth intervals along the core (see Table 1, "N" series, and Table S1 for precise depths), in order to be representative of both the dry and the wet seasons. The samples were filtered using PTFE Millipore© membranes (0.45 μm pore size, 11.3 mm diameter) previously rinsed in an ultra-pure 5% solution of bi-distilled $HNO_3$— according to the procedures adopted by Baccolo et al. (2015). The filtration took place in an ISO 5 Class laminar flow bench. Two blank membranes, that underwent the same cleaning procedures, were prepared by filtering 300 mL of MilliQ (Millipore©) water. For calibration and quality control, we used certified solid standards: USGS AGV2 (ground andesite), USGS BCR2 (ground basalt), NIST SRM 2709a (San Joaquin soil) and NIST SRM 2710a (Montana soil). In addition, standard acid solutions for each analyzed element were prepared with concentrations in the order of μg g$^{-1}$. Blanks for the empty flask and for the ultra-pure acid solution used to prepare the liquid standards were also measured. Samples, standards and blanks were irradiated at the Applied Nuclear Energy Laboratory (LENA, University of Pavia, Italy) by a Triga Mark II reactor of 250 kW. The "Lazy Susan" channel, neutron flux equal to $2.40 \pm 0.24$ x $10^{12}$ s$^{-1}$cm$^{-1}$, was used to identify Ce, Cs, Eu, Hf, La, Sc, Sm, Th, and Yb. Samples were successively transferred at the Radioactivity Laboratory of the University of Milano-Bicocca, in order to acquire gamma spectra by means of a high-purity Germanium detector HpGe (ORTEC, GWL series), following the standardized procedure developed for low-background INAA (Baccolo et al., 2016).

The masses of the elements in each sample were determined by comparing spectra related to standards and samples (Baccolo et al., 2016). In order to compare different spectra, the time of acquisition, the radionuclide decay constant, the cooling time and a factor considering radioactive decay during the acquisition were kept into account. The detection limits were calculated considering three times the standard deviation of the blank signal. The uncertainties for each element were calculated based on the mass measurements, the adjustment for the spectrum, the subtraction of the blanks and the standard concentration uncertainties. Errors for the elemental concentrations in our samples ranged from 3% for La to 17% for Cs, and the detection limits ranged from 0.1 μg per gram of dust for Sm to 7 μg g$^{-1}$ for Ce (Table S2). Full analytical details can be found in Baccolo et al. (2016).

The Enrichment Factor (EF) normalization was calculated for each element considering as a reference the mean composition

of the upper continental crust (UCC) (Rudnick and Gao, 2003). Scandium (Sc) was chosen as the crustal reference element following Eq. (1):

$$EF(x) = \frac{\left(\frac{[X]}{[Sc]}\right)_{sample}}{\left(\frac{[X]}{[Sc]}\right)_{UCC}} \tag{1}$$

Scandium was chosen as reference element because it is poorly affected by processes altering its mobility in hosting minerals, and its biogeochemical cycle is almost unaffected by anthropogenic activities (Sen and Peucker-Ehrenbrink, 2012). In addition, Sc is highly correlated with other lithogenic elements, such as Ce (r = 0.997), used by Eichler et al. (2015) as a crustal
reference for Nevado Illimani samples, and La (r = 0.989). The choice of Sc has been also determined by its easy and precise determination through INAA.

## 2.4   Micro-Raman Spectroscopy

We used single-grain Raman spectroscopy to identify mineralogy of dust particles having a diameter smaller than 5 μm. This,
because this kind of analysis was carried out for provenance purposes, thus considering particles expected to travel over longer distances. A set of four samples (see Table 1, "R" series, and Table S1) was prepared, following the procedure described in previous studies (Delmonte et al., 2017; Paleari et al., 2019) specifically developed for small dust grains. Two samples are representative of mineral dust deposited in the dry season (high dust concentration) whereas two represent dust from the wet season (low dust concentration, or "background"). Measurements were performed by using an InVia Renishaw micro-Raman
spectrometer (Nd YAG laser source, $\lambda = 532$ nm) available at the Laboratory for Provenance Studies (UNIMIB). We identified the mineralogy of more than 630 grains, excluding organic particles possibly related to contamination and particles with an undetermined spectrum or with no signal.

## 2.5   Stable Isotope and Ion Chromatography analyses

The dust analyses described above used one quarter of the longitudinally-cut firn core. A second quarter was shipped in a frozen state to the Climate Change Institute (CCI, University of Maine, USA) for ion chromatography (IC) and water stable isotope analysis.

At the CCI, in a cold room set at -20 °C, we cut longitudinal sections of the core with a vertical band saw to separate an inner and an outer part. The latter was sampled by transverse cuts approximately every 12 cm using a stainless-steel hand saw
(resulting in 190 samples) and stored in plastic bottles for stable isotope ratio determination. Decontamination of the inner part was performed by scraping with a clean ceramic knife under a laminar flow HEPA bench inside the cold room. Then, the decontaminated inner part was sampled for IC analysis by a continuous melter system (Osterberg et al., 2006) also in an ISO 6 Class clean room. The mean sample resolution was 3 cm, resulting in 767 samples. We measured $Ca^{2+}$ concentration using

a ThermoScientific™Dionex™ Ion Chromatograph ICS-6000 analytical system fitted with suppressed conductivity detectors, and a Dionex AS-HV autosampler. The method detection limit (MDL) was defined as 3 times the standard deviation of the blank samples (MilliQ water, 10 blank samples). The detection limit for $Ca^{2+}$ was 21.05 μg $L^{-1}$.

The $\delta D$ and the $\delta^{18}O$ were determined using a Picarro L2130-i wavelength-scanned cavity ring-down spectroscopy instrument (Picarro Inc., USA) with a precision of 0.1‰.

## 2.6 Correlation evaluation

The correlation between GPPnb and $\delta D$ was examined using their random components, which were obtained by extracting both their seasonality and outliers. The annual cycles were removed by subtracting the averages for each season, which are defined in Sect. 3.1 as "wet", "dry", and "transition". Based on the statistical random distribution of GPPnb and $\delta D$, values above 3 standard deviations were considered to be outliers. As the resulted GPPnb random component was not normally distributed, a Spearman's rank correlation was used to asses the correlation (implemented in the Scipy library of numerical routines for the Python programming language, Virtanen et al., 2020). The Confidence Interval (CI) was obtained using a block bootstrap resampling method, following Mudelsee (2014). This method produces simulated time series of the same length, and calculates correlation coefficients for each simulation. By resampling blocks of the random components data, persistence over the block length was preserved. An optimal block length was calculated considering a first order autoregressive persistence model, where a realization of the random process depends on just the value of a time step earlier. The CI was calculated from 2000 bootstrap simulations (run by the Recombinator Python package, https://pypi.org/project/recombinator/), and then obtained using Fisher's transformation.

## 3 Results and discussion

### 3.1 Seasonal variability of proxies and firn core chronology

We established a chronology for the Nevado Illimani firn core based on annual layer counting (ALC), considering the pronounced seasonal oscillation of dust concentration, calcium and water stable isotopes (Fig. 2). Dust concentration variations, which are recognized for being useful for ALC in tropical and continental ice cores (Ramirez et al., 2003; Kutuzov et al., 2019), span about two-orders of magnitude between the summer and the winter. Dust concentration varies from ~2,000 particles $mL^{-1}$ (hereafter part. $mL^{-1}$) during the wetter season, to ~10,000 part. $mL^{-1}$ during the dryer season (median values). The two size distributions shown in Fig. S2, illustrate this variability. When considering extreme values, the variation range exceeds three orders of magnitude, being the lowest concentration during the wet season 150 part. $mL^{-1}$ and the highest one during the dry season 140,000 part. $mL^{-1}$. Our results are in agreement with average dust concentrations from Quelccaya ice cap during the 20th century, ~10,000 part. $mL^{-1}$ and ~25,000 part. $mL^{-1}$ for the size ranges of 1.6–16 μm and 0.6–20 μm,

respectively (Thompson et al., 1986, 2013). By considering just the giant particles we also observed a seasonal pattern, with median concentrations of 15 part. $mL^{-1}$ during the wet season and 30 part. $mL^{-1}$ during the dry season. The well-defined oscillatory pattern of dust concentration variability reflects the extreme seasonality of precipitation over both local and regional dust sources, and the succession of dry and wet conditions. Sublimation has a limited influence to this seasonality (Ginot et al., 2002).

Dust concentration is in accordance with the $Ca^{2+}$ record and also with literature studies (Knüsel et al., 2005). However, both records show differences, in particular during the dry season when they are not significantly correlated at the 95% level. Considering our high temporal sampling resolution, this might be associated with slight changes in dust mineralogy, possibly affecting the amount of calcium to be solubilized. Ionic calcium can be primarily associated to calcium sulphate ($CaSO_4$) and/or calcium carbonate ($CaCO_3$) (Kutuzov et al., 2019). Because scarcity of calcium carbonates was revealed by miner-
alogical analyses (Fig. 4, see below), we argue that most of the ionic calcium observed in firn samples is present as a soluble species, probably $CaSO_4$, and not detectable through Raman spectroscopy on single insoluble particles. However, we consider the possibility of calcium carbonate depletion due to scavenging during dust transport and/or dissolution during the melting of the samples, as discussed by Wu et al. (2016) based on ice core samples from the Tibetan Plateau. In addition, we cannot exclude that Ca-bearing aerosols might have been initially a mixture of pure gypsum and calcium carbonates that successively
reacted with atmospheric $H_2SO_4$, in the atmosphere or within the snow-pack as the result of post-depositional processes (Röthlisberger et al., 2000; Iizuka et al., 2008).

The regular succession of dry dusty periods and wet periods can be associated with the seasonal onset and decay of the Bolivian High, a high pressure system which is well developed and centered over Bolivia (Lenters and Cook, 1997). When the Bolivian High is particularly strong and displaced southward of its climatological position, easterly flow in the high troposphere
is enhanced as well as moisture advection from the interior of the continent to the Altiplano. This moisture transport from the Amazon Basin toward the Altiplano induces a notable amount of precipitation over the Altiplano (wet season), associated with strong summer convection. The relatively low dust concentration found in the Illimani snow during the summer period is therefore related to particle dilution in the snowpack because of increased precipitation and reduced regional dust mobilization deriving from wetter soil conditions. Conversely, during winter (JJA) months, conditions over the Altiplano are typically dry,
leading to higher dust availability. At that time of the year, the winter westerly flow over the entire region promotes eastward dust transport towards the Nevado Illimani, leading to significant higher dust deposition in the firn layers representing — on average — about 85% of the total annual dust particles there deposited.

Seasonal variations of the water stable isotopes in snow precipitated over the Andes are also useful for dating. However, the Andean isotopic signal led to divergent interpretations (Vimeux et al., 2009). Whereas in polar ice cores the water isotopic
signature is chiefly related to temperature (Uemura et al., 2012), the isotopic composition of tropical precipitation can be affected by a larger number of factors (Hoffmann et al., 2003). It is well known that the so called "amount effect" leads to an anti-correlation between the amount of precipitation and the proportion of heavier isotopes in the precipitation. This effect is in turn related to an ensemble of physical and microphysical processes producing a robust signal on the isotopic composition of precipitation (Dansgaard, 1964; Vuille et al., 2003; Risi et al., 2008). In this context, deep atmospheric convection also plays

a role on stable isotope composition (Vimeux et al., 2005). Along the Zongo Valley (Fig. 1, located nearby Nevado Illimani), in particular during the summer season, the cumulative rainfall along air mass trajectory is a second order parameter in the control of isotopic depletion, being it primarily modulated by regional convective activity (Vimeux et al., 2011). In agreement, modelling studies (e.g., Bony et al., 2008; Risi et al., 2008) reveal that the stronger the convective activity during a particular event, the higher the total amount of precipitation and thus the more depleted the isotopic composition of precipitation. In

addition, satellite data (Samuels-Crow et al., 2014) reveal that during the summer season the isotopic composition of water vapor strongly depends on convective activity. These observations lead the authors to conclude that the isotopic composition of snow from the tropical Andes mainly reflects tropical convection. Convective precipitation over the Bolivian Altiplano is enhanced during the wet summer season, leading to the emergence of clear seasonal oscillations in the stable isotope records of Nevado Illimani firn core which can be used for ALC and to develop a chronology.

Considering the pronounced seasonal changes in dust concentration, $Ca^{2+}$ and stable isotopes, it is possible to assign to the base of the Illimani firn core an age corresponding to the beginning of 1999 AD. The firn record thus covers the 18-years period from early 2017 to early 1999 and the average accumulation rate can be estimated in the order of approximately 750 mm of water equivalent per year, slightly higher than the one inferred by Knüsel et al. (2003). In addition, data was classified by season following the procedures in (Correia et al., 2003) by individually grouping the samples into three categories ("dry",

"wet", "transition") according to concentration levels of dust, $Ca^{2+}$ and stable isotopes. The samples belonging to the depth intervals in the red (blue) areas of Fig. 2 were classified as dry (wet) season samples. All other samples were classified as belonging to the transition season.

     Interestingly, we note a close correspondence between the variability of stable isotopes and the proportion of giant particles in firn (Fig. 3): oscillations of the stable isotope record ($\delta$D) closely follow the percentage of giant dust particles (GPPnb). During

the dry season, giant particles are proportionally less abundant (average GPPnb 0.5%) whereas the isotopic composition of snow is less negative (average $-113‰$ for $\delta$D; $-15‰$ for $\delta^{18}$O). Conversely, during the wet season when giant dust particles are at their annual maximum (average GPPnb 1%) the isotopic composition of snow is more depleted ($-141‰$ for $\delta$D, $-18‰$ for $\delta^{18}$O), reaching its minimum. We found a significant correlation between GPPnb and $\delta$D at the 95% level (r $= -0.53$, p $< 0.001$, n $= 263$). The CI for r was $[-0.35, -0.67]$, being inside the 95% level significance range. Considering the absolute

concentrations of dust and giant particles, they showed weaker correlations with $\delta$D (0.14 $[-0.03, 0.31]$, $-0.11$ $[-0.28, 0.08]$, respectively). Only this first correlation is significant at the 95% level.

## 3.2    Dust provenance: mineralogy and geochemistry

     The mineralogical composition of fine dust ($<5$ μm) deposited onto the Illimani firn layers reveals that the most abundant

mineral phases are quartz, feldspars (alkali feldspars and plagioclase) and phyllosilicates (Fig. 4, Table S3). Phyllosilicates are mainly represented by muscovite-illite (and/or smectite, hardly distinguishable from illite by their Raman spectra) and secondarily by kaolinite (representing 7.5% in the dry season and 1.5% in the wet season). Altogether, quartz, feldspars and phyllosilicates account for 75–78% of mineral particles during both the wet and the dry season, but phyllosilicates are par-

ticularly abundant during the wet season, when they represent approximately 44% of minerals (Fig. 4a). We believe that the increased abundance of muscovite-illite during the wet season is related to different depositional regimes. During the dry winter, aerodynamic platy-like phyllosilicates can remain in the atmosphere for longer periods and are only partially deposited. On the contrary, during the wet summer strong scavenging is associated to heavy precipitations at Nevado Illimani (Bonnaveira, 2004), enhancing the removal of mineral particles from the atmosphere, including phyllosilicates.

Titanium oxides and iron oxides/hydroxides are present in all samples. Hematite is twice as abundance as goethite. This is typical in regions dominated by arid conditions or where a prolonged warm and dry season is followed by a shorter and wetter period (Journet et al., 2014). Accessory minerals include carbonate and tourmaline, and very rare pyroxenes (Table S3). Such a mineralogical composition is coherent with the felsic to intermediate plutonic volcanic source rocks, suggesting that most of the dust deposited at Nevado Illimani has a local/regional provenance, both in the wet and in the dry seasons.

Low-background INAA analyses allowed determining the EFs for different rare earth elements (REEs), which are non-mobile and therefore widely used as provenance tracers (McLennan, 1989; Moreno et al., 2006; Gabrielli et al., 2010). In Fig. 5a the $Yb/La$ and the $Eu/Sm$ elemental ratios are used to compare dust samples retrieved from Nevado Illimani firn, and literature data concerning geological samples from the Altiplano-Puna Volcanic Complex (Ort et al., 1996; Lindsay et al., 2001) and potential source areas (PSAs) in South America (Gaiero et al., 2004, 2013). The $Yb/La$ ratio can be used to appreciate whether heavy and light REEs are enriched or depleted with respect to each other, whereas the $Eu/Sm$ ratio is a proxy for the europium anomaly, usually calculated considering $Gd$, not detected by our analytical method. The comparison reveals that Nevado Illimani dust has a composition similar to APVC crystal-rich ignimbrites, pointing to a correspondence with samples from the Northern Puna region, and not with samples from the salt lakes present in the Altiplano (Uyuni and Coipasa salars). These pieces of evidence agree with previous analyses of strontium and neodymium isotopes on Nevado Illimani ice core dust (Delmonte et al., 2010) and with the geochemical signature of sources in the Altiplano (Gili et al., 2017), supporting the hypothesis that dust deposited at Nevado Illimani is sourced by sediments present in Southern Altiplano-Northen Puna areas.

The EFs of $Ce$, $La$, $Sm$, $Eu$ and $Yb$ are similar between samples with higher percentage of giant particles (GPPnb > 1%, characteristic from the wet season) and samples with background GPPnb (Fig. 5b). This corroborates what was observed in relation to the mineralogical composition of the samples from the wet and dry seasons. However, two important exceptions to this pattern occur in relation to $Hf$ and $Cs$. Samples with high GPPnb show anomalous enrichment of $Hf$ when compared to background samples. A similar feature was also observed in Saharan dust samples (Castillo et al., 2008) and attributed to the presence of detrital zircon $(Zr, Hf)SiO_4$ in samples showing the coarsest grain sizes. Following Vlastelic et al. (2015), the $Hf$ enrichment observed in samples where the GPPnb is higher may be related to the presence of a few silt-sized zircon grains, which in turn would require high energy (turbulence) to lift and keep them suspended in the atmosphere. However, zircon grains were not detected by Raman Spectroscopy in this study. The mineralogical analysis indicated a greater abundance of phyllosilicates in the dust deposited during the wet summer season. Thus, the $Hf$ enrichment in these samples might be related to tiny zircon inclusions within phyllosilicate particles. In accordance, the $Cs$ enrichment in samples with a higher GPPnb may also be related to a greater abudance of phyllosilicates during the summer. In the interlayer sites of illite-muscovite minerals the $Cs-K$ exchange is very common (Cremers et al., 1988; Rosso et al., 2001). We conclude that the geochemical variability

between samples with high and background GPPnb seems to be mainly related to the variability of phyllosilicate concentration. In turn, the variability of these minerals is related to scavenging, and therefore precipitation.

### 3.3 Relationship between the giant particles and deep convection

The absolute concentration of dust in firn and ice cores depend on many factors including the snow accumulation rate, the dust source strength (which includes soil aridity/wetness, vegetation cover, and any other factor influencing the quantity of particles available for deflation), as well as transport processes, which also affect the residence time of particles in the atmosphere (mainly in the case of long-range transport) (Delmonte et al., 2004). In the case of Illimani, where dust is mostly locally and regionally-sourced (Sect. 3.2), we believe that dust concentration is primarily modulated by the seasonally-varying source strength, mainly depending on source aridity and humidity, and by accumulation rate. Indeed we observe a pronounced decrease in dust concentration during the wet season, when convective activity reduces the source strength and increases the snow accumulation. Interestingly, during the wet season the relative number of giant particles in the firn core increases (Fig. 3). In addition, the variability of both GPPnb and $\delta$D during the wet season shows a significant correlation (Fig. 6, r $= -0.69$ [$-0.58$, $-0.79$], p $< 0.001$, n $= 123$). As a more depleted isotopic composition of precipitation is caused by a more intense summer convection, this also seems to lead to a higher GPPnb (data located in the bottom right corner of Fig. 6).

Convective activity is known to significantly affect the isotopic composition of tropical precipitation. Intense regional convection leads to more isotopically-depleted precipitation (e.g., Risi et al., 2008). A proposed mechanism is that convective downdraft promotes the subsidence of higher level water vapor, causing isotopic depletion of low level vapor crossing the Eastern Cordillera of the Central Andes (Vimeux et al., 2011). This downdraft is generated by the cooling of air due to the reevaporation of the falling precipitation, which is favored by the often dry (unsaturated) conditions over the Central Andes. Conversely, the $\delta$D variability during the winter responds mainly to the intense reevaporation processes occuring in that dry atmosphere, which features a low convective activity (Vimeux et al., 2011). Convective downdrafts, as observed during the wet season, are often associated with density currents, offering an efficient mechanism for dust lifting (Flamant et al., 2007). Indeed, the leading edge of the density current is characterized by strong turbulent winds that can mobilize dust and mix it through a deep layer (Knippertz et al., 2007). In accordance, events of giant dust particles suspension and transport, as detected by aircraft measurements in north Africa, were related to the occurrence of convective systems (Ryder et al., 2013). Although we have not observed any significant correlation between $\delta$D and the absolute concentration of giant particles, we must consider that the effect of convection on giant particle concentration might be twofold. During the summer, it favors turbulent conditions for the suspension of giant particles, but also provides heavy precipitation, reducing the source strength in the sources of giant particles and increasing the accumulation. In fact, the absolute concentration of giant particles is lower during the wet season than during the dry season by a factor of 2 (Fig. 2). Thus, the major source areas of giant particles, probably local (but this would require a specific provenance study for giant particles), might be strengthned during dry conditions. Our finding is that summer deep convective activity leads to a lower dust concentration in the firn core, and also to a relatively lower reduction in the concentration of giant particles. Therefore, the relative number of giant particles on the Nevado Illimani

glacier can be reasonably used as a proxy for deep summer convective precipitation. Given the size of these particles and the dust geochemical/mineralogical fingerprint, we confidently associate the giant particles to local/regional convective activity.

In order to test the hypothesis of a relationship between giant particles and convective precipitation, we analyzed monthly precipitation from five meteorological stations located in the central Andes (Fig. 1), and monthly outgoing longwave radiation (OLR) centered at 17.5°S, 70°W. Low OLR values correspond to cold and high clouds which denote enhanced convection. It is estimated that deep convection provides 65% of the precipitation over this region, as the orographic lifting of moisture from the Amazon basin through Andes trigger condensation, latent heat release, and strong convective updrafts during the summer

(Insel et al., 2010). In agreement, OLR shows strong negative correlations with regional rainfall observations over the Bolivian Altiplano (Garreaud and Aceituno, 2001). Furthermore, both OLR and precipitation data provided similar results when linking $\delta$D and regional convection at the Zongo Valley (Vimeux et al., 2011). Precipitation data was provided by SENAMHI, Bolivia (www.senamhi.gob.bo/sismet), whereas monthly OLR data on a 2.5° x 2.5° grid box (Liebmann and Smith, 1996) was obtained from NOAA/OAR/ESRL PSD, Boulder, Colorado, USA (https://www.esrl.noaa.gov/psd/). These datasets had their

annual cycle removed by subtracting the monthly averages over the period 1999–2017. Then, they were resampled into DJF (December to February) and JJA (June to August) time series and compared with the random components of our seasonally resolved GPPnb series. For each wet and dry season, defined according to dust concentration, $Ca^{2+}$, and $\delta$D records (Sect. 3.1), a mean GPPnb was obtained.

In Table 2 we show the results of a Spearman correlation analysis between seasonal GPPnb and meteorological data. No sig-

375 nificant correlation at the 95% level was observed during the dry season, therefore, only the wet season correlations are shown. Table 2 clearly show that during wet season GPPnb is positively correlated (at 95% level) with DJF precipitation at Patacamaya (17.2°S, 67.9°W, 4498 m a.s.l.). Rainfall variability over the Altiplano is strongly dependent on the intensity of the moisture transport over the eastern slope of the Eastern Cordillera, but also depends on the local amount of near-surface water vapor (Garreaud, 2000). This later responds to the complex topography of the Central Andes, resulting in differences between the

380 precipitation records of Altiplano's meteorological stations (Aceituno, 1996). Thus, it is expected that only precipitation data from stations in the closest vicinity of the Nevado Illimani show good correspondence with glaciological data. In agreement, Knüsel et al. (2005) observed that the Patacamaya precipitation record was better correlated (compared to El Alto) with the dust related ions record from Nevado Illimani, suggesting that the precipitation regime in the area south of Nevado Illimani influences its dust record.

As convective clouds are formed over the Eastern Cordillera, the reevaporation of precipitation falling through the dry atmospheric boundary layer on its western slope leads to a downward flow of cold air over the highly complex terrain. This is driven by differences in density in relation to the environment, similar to the mechanism proposed by Knippertz et al. (2007). The leading edge of this density current is characterized by strong turbulent winds, suggesting giant dust particles mobilization over areas in the vicinity of the Nevado Illimani, where these particles are also deposited. In accrodance, GPPnb is negatively

correlated with the DJF OLR centered over the Altiplano (Table 2), indicating that deep convection increases giant particle entrainment and suspension, humidity and precipitation over the region. Curiously, we found no significant correlations between GPPnb and wind speeds at the meteorological stations. This might be due to the short lifetimes of the density currents related

to convection, observed to be in the order of a few hours (Knippertz et al., 2007). Considering our seasonal resolution analysis, we suggest that these high turbulence events had a low influence on the mean DJF wind speed.

We conclude that the more intense is summer convection, the higher is the relative number of giant dust particles suspended in the atmosphere and the more depleted is the $\delta$D. Differently from the wet season, when the major control of $\delta$D variability in Zongo (and therefore probably in Illimani) is the progressive depletion of water vapor by unsaturated convective downdrafts, the $\delta$D variability of the winter responds mainly to the intense reevaporation processes that occur in a dry atmosphere, with a low convective activity (Vimeux et al., 2011). The rare winter convection seems also to have a low influence in GPPnb vari-

ability, as indicated by its lack of significant correlations with both JJA precipitation and OLR.

Fig. 6 shows that over the 18-years period analyzed in this work, the summer seasons of 2000–2001 and 2010–2011 showed intense levels of convection (considering both GPPnb and $\delta$D). Both correspond to La Niña periods, as indicated by their DJF Oceanic Niño Index (ONI) of $-0.7$ and $-1.4$, respectively. It is well known that the El Niño-Southern Oscillation phenomenon has a significant impact on climate over the Altiplano, especially during the summer season. In particular, meteorological data

show that La Niña conditions intensify the meridional pressure gradient on the northern side of the Bolivian High, leading to stronger high troposphere easterly winds, increased eastward upslope flow and enhanced moisture transport (Garreaud, 1999; Vuille, 1999). However, the strong DJF La Niña events of 1999-2000 (ONI = -1.7), and 2007-2008 (-1.6) do not show higher GPPnb or more depleted $\delta$D compared to other values of the wet season (Fig. 6). We believe this was due to competing mechanisms controlling moisture transport from the Amazon basin to the Altiplano. In addition to the role played by the upper

troposphere easterly winds, the meridional circulation between the tropical North Atlantic Ocean and the western tropical South America also influences the DJF precipitation over the Central Andes, especially in the last two decades (Segura et al., 2020). Evidence based on reanalysis data indicates that when this meridional circulation is enhanced, the atmospheric stability between the mid and the upper troposphere over the Altiplano is reduced, resulting in increased moisture transport from the Amazon basin (Segura et al., 2020). Thus, we propose a new approach for future studies in tropical Andean glaciers based

on giant particles and stable isotopes of snow. This can be used as a complement to a number of other climate proxies and modeling experiments, providing insights into past atmospheric circulation over tropical South America.

*Data availability.* Dust (concentration, grain size, geochemistry and mineralogy), stable water isotopes and calcium data can be made available for scientific purposes upon request to the authors (contact filipelindau@hotmail.com, jefferson.simoes@ufrgs.br or barbara.delmonte@unimib.it).

*Author contributions.* FGLL, JCS, BD, GB wrote the original manuscript. JCS and BD designed the research. PG designed and led the drilling campaign. FGLL and PG sampled the core. FGLL, BD, GB, CIP and EDS conducted dust analyses. EK and DSI carried out the ionic and the isotopic measurements, respectively. BD, EG, SA, GB and CIP advice on data collection and interpretation. VM, CIP, PG, EG and SA provided comments to the original manuscript. VM, EG and SA provided analytical resources.

*Competing interests.* The authors declare that they have no conflict of interest.

*Acknowledgements.* We thank the drillers S. Kutuzov, L. Piard, B. Jourdain, all the operation team and the support of IRD office in Bolivia. Operations at Illimani were part of the Ice Memory project, financed by IRD, Université Grenoble Alpes, CNRS, IPEV and UMSA and by a sponsorship from the Université Grenoble Alpes Foundation. This research was partially funded by NSF project 1600018, by the Brazilian CAPES, project 88887.136384/2017-00 and a research grant from the Brazilian National Council for Scientific and Technological Development (CNPq 465680/2014-3). F.G.L. Lindau thanks CNPq for his scholarship (Processes 141013/2015-0 and 200496/2017-4). We
are grateful to two anonymous reviewers and the Editor for their suggestions for improving the manuscript.

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

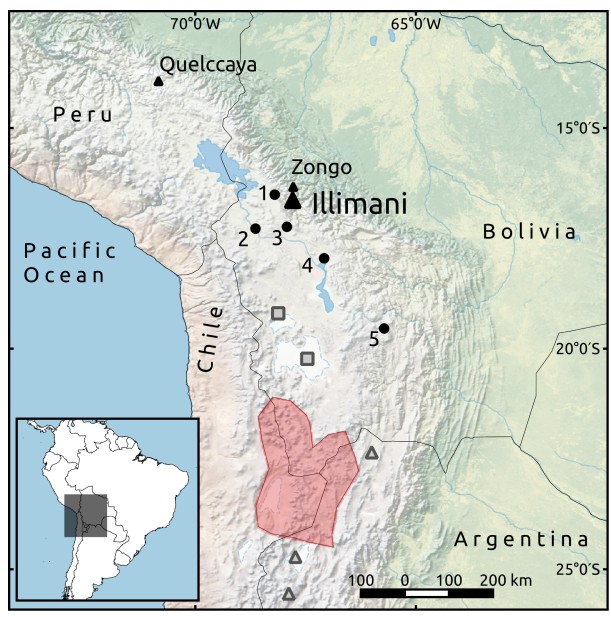

**Figure 1.** Location of the Nevado Illimani, the Zongo Valley and the Quelccaya ice cap. The numbers indicate the location of the meteorological stations used for comparison with our results: 1 – El Alto; 2 – Calacoto; 3 – Patacamaya; 4 – Oruro and 5 – Potosi. The red area indicates the Altiplano-Puna Volcanic Complex (Lindsay et al., 2001). The gray squares and gray triangles denote to potential dust source areas in the salars of the Altiplano and in the Puna, respectively (Gaiero et al., 2013). The land basemap was derived from satellite data (Natural Earth I with Shaded Relief from http://www.naturalearthdata.com).

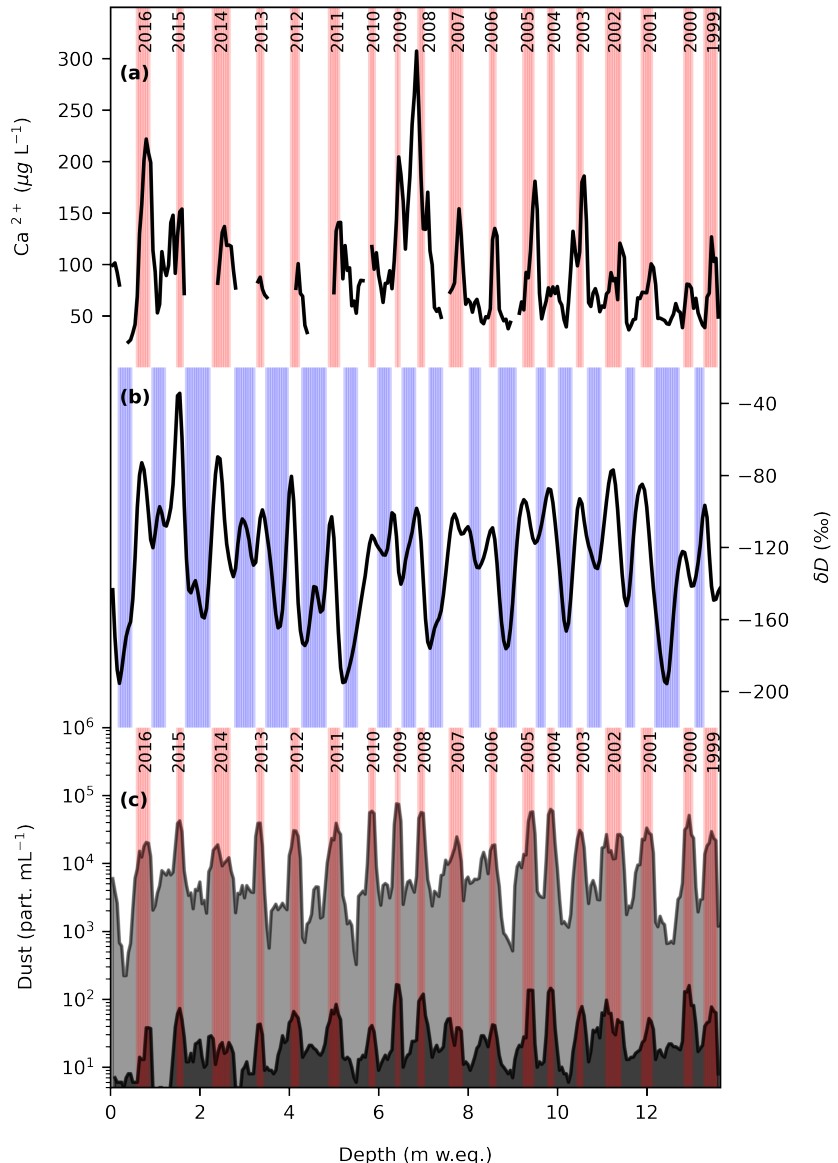

**Figure 2.** Dating of the Nevado Illimani firn core by annual layer counting (ALC) based on different proxies discussed in the text: (a) ionic Calcium, (b) δD, and (c) total and giant dust particles concentrations (light and dark gray, respectively, both are in logarithmic scale). Red and blue shaded vertical bands correspond respectively to the dry and wet season for each calendar year. All data are reported as 3-point running average of data re-sampled at 0.05 mw.eq.

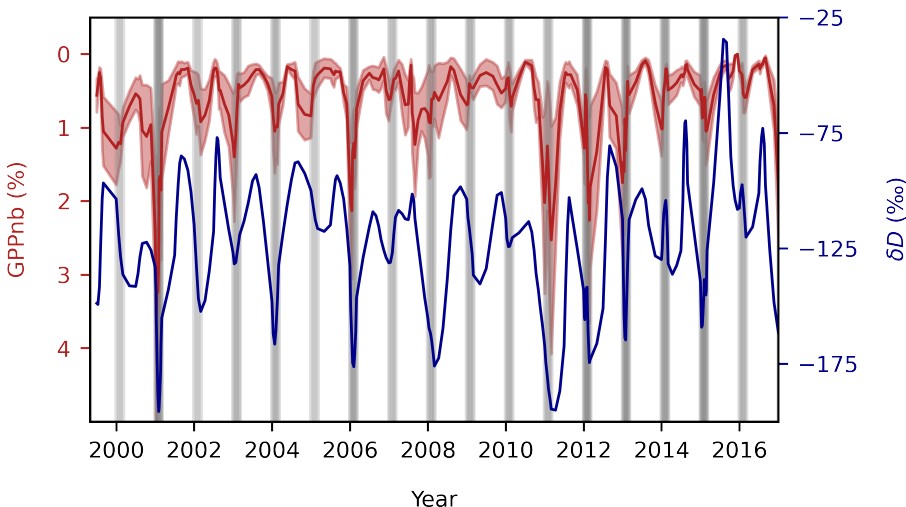

**Figure 3.** Relationship over the 18-years record between the percentage of giant particles with respect to the total dust particles number (GPPnb, reverse scale) and the $\delta$D. Gray shaded vertical bands correspond to the wet season for each calendar year. Uncertainties for each GPPnb value (expressed by the red shaded area) are relative to the standard deviation between Coulter Counter measurements. All data are reported as 3-point running average of the data previously re-sampled at 0.05 m w.eq.

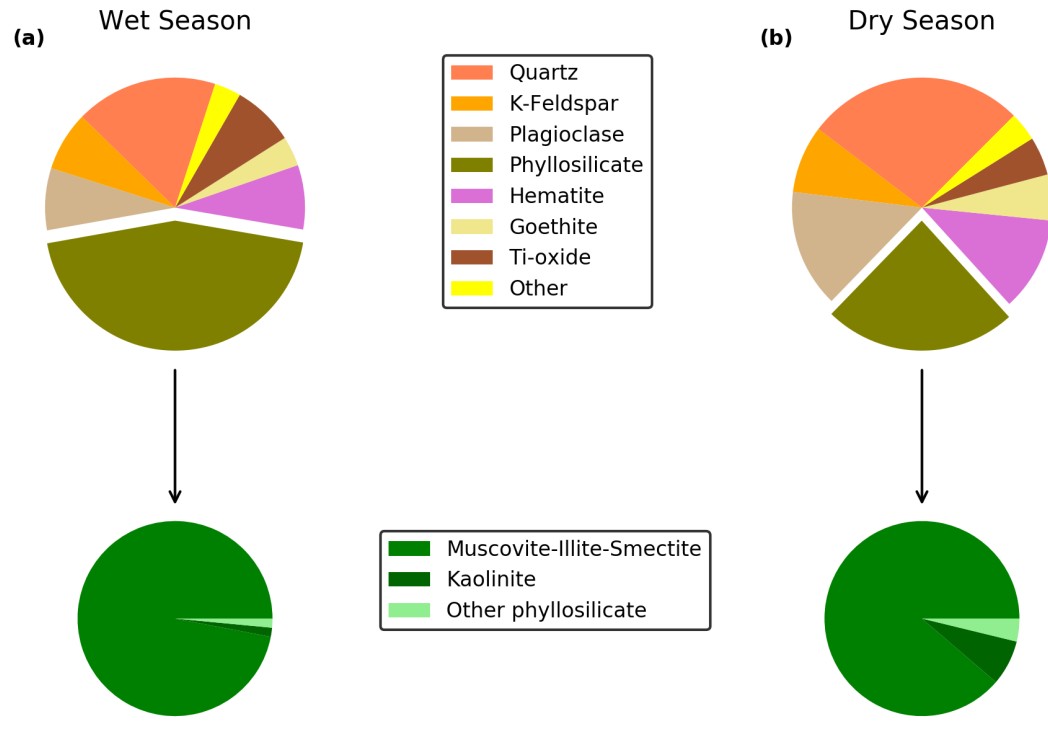

**Figure 4.** Changes in dust mineralogy between (a) the wet and (b) the dry seasons. The lower plots highlight the mineralogy of the phyllosilicates.

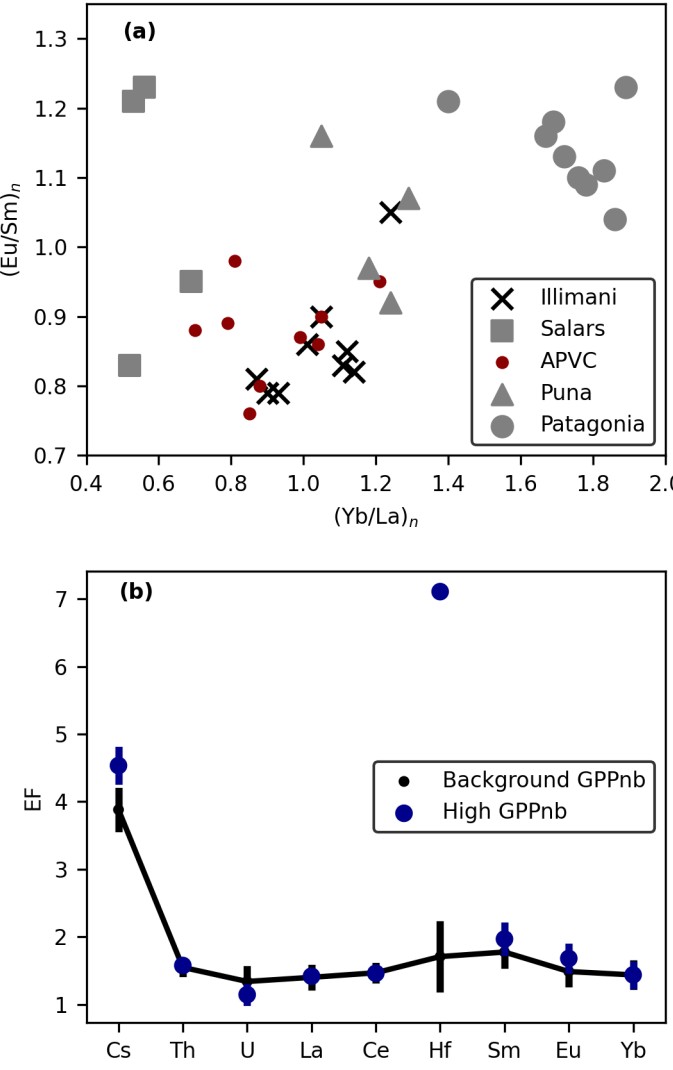

**Figure 5.** Geochemical signature of Nevado Illimani firn samples. (a) Relationship between the REE (normalized considering the UCC composition Rudnick and Gao, 2003) from the Nevado Illimani firn core (this work) and sediments/soils from potential dust sources (corresponding to < 63 μm grain size for top soils). Data from Northern Puna, and Uyuni and Coipasa salars are from Gaiero et al. (2013). Data from Patagonia are from Gaiero et al. (2004). Data for the APVC refer to geological samples from Lindsay et al. (2001) and Ort et al. (1996). (b) Enrichment Factors (EF) for different elements and standard deviations. Samples with a high GPPnb (blue circles) show anomalous enrichment for Hf and to a lesser extent for Cs (see text).

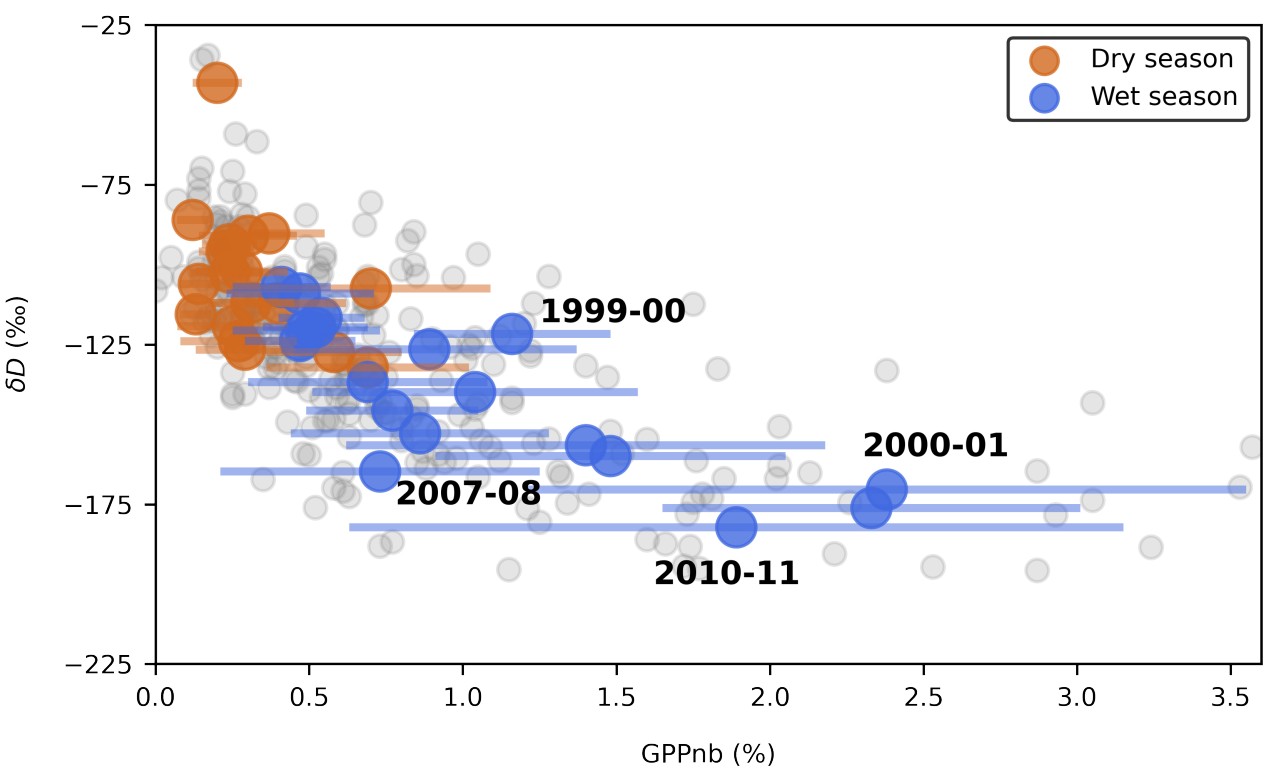

**Figure 6.** Seasonal mean GPPnb and $\delta$D for the dry seasons (orange circle) and the wet season (blue circle). Error bars (horizontal bands) for GPPnb are based on the mean relative standard deviation for the samples integrating each season. Light gray dots on background are raw data. The La Niña events discussed in the text (1999–00, 2000–01, 2007–08, and 2010–11) are reported.

**Table 1.** Characterization of the Nevado Illimani samples analyzed for elemental (N1 to N10) and mineralogical (R1 to R4) composition.

| Sample | Year | Season | Dust (part. $mL^{-1}$) | GPPnb (%) |
|--------|------|--------|------------------------|-----------|
| N1 | 2016 | Dry | 14,737 | 0.1 |
| N2 | 2015 – 2016 | Wet | 7,283 | 0.2 |
| N3 | 2013 – 2014 | Wet – Dry | 6,340 | 0.3 |
| N4 | 2012 | Dry | 30,405 | 0.2 |
| N5 | 2010 – 2011 | Wet | 2,605 | 1.3 |
| N6 | 2008 | Dry | 25,816 | 0.7 |
| N7 | 2007 – 2008 | Wet – Dry | 6,424 | 1.0 |
| N8 | 2003 | Dry | 29,923 | 0.2 |
| N9 | 2002 | Dry | 21,057 | 0.2 |
| N10 | 2000 – 2001 | Wet | 4,180 | 1.9 |
| R1 | 2010 | Dry | 17,409 | 0.1 |
| R2 | 2009 – 2010 | Wet | 8,234 | 0.2 |
| R3 | 2004 | Dry | 86,918 | 0.2 |
| R4 | 2003 – 2004 | Wet | 2,303 | 0.9 |

**Table 2.** Spearman's correlations, between Giant Particles Percentage (GPPnb), rainfall observations and outgoing longwave radiation (OLR). All data refers to the wet season (December–January–February). The annual cycle of the meteorological data was removed by subtracting the monthly means. Outliers were also removed. Correlation coefficients (r) that are significant at the 95% level are shown in bold. The P-value for each correlation, as well as, the number of data points (n) are also shown.

|  | Wet Season GPPnb | | |
|---|---|---|---|
|  | r | P-value | n |
| El Alto | 0.22 | 0.390 | 18 |
| Calacoto | 0.07 | 0.798 | 18 |
| Patacamaya | **0.80** | <0.001 | 16 |
| Oruro | **0.48** | 0.044 | 18 |
| Potosi | 0.07 | 0.785 | 18 |
| OLR | **-0.70** | 0.001 | 18 |