# Peer review of "Giant dust particles at Nevado Illimani: a proxy of summertime deep convection over the Bolivian Altiplano"

_The Cryosphere, 2020_

## Referee Comment (RC1) · Anonymous Referee #1 · 13 May 2020

This manuscript presents a shallow ice core record of mineral dust concentration and composition from Illimani, Bolivia, covering the time period 1999 to 2017. Generally, it is well written and structured and mostly scientifically sound (see comments below). The authors propose the relative proportion of giant dust particles as proxy for summer convective activity, which is an interesting new approach. The topic is within the scope of TC and the manuscript deserves publication, after taking into account the comments and suggestions listed in the following.

General comments

The correlation between giant particles and $\delta$D, considered as proxy for convection,

is based on the relative mass proportion of giant particles. Considering the fact that determining giant particles in a liquid sample is tricky, because they tend to settle, you need to substantiate the robustness of your results better. Specify if the given mean standard deviation between the two measurements with the coulter counter (lower than 3%) applies also to the giant particles alone. Show examples of the size distributions for the wet and dry season. Since total particle mass concentrations are low during the wet season (when you observe the correlation), the relative mass proportion is the ratio of two small numbers, probably having a large uncertainty. Add uncertainty bars and show in addition the record of mass or number of giant particles for every sample. From the presented data it is unclear if the number or mass of giant particles also has a seasonal or any other variability or not.

Before being lifted up by convection, dust particles need to be mobilized from the ground, which requires strong wind (advection). Have you checked wind speeds in the dust source areas? Dust source areas are located SE of Illimani, whereas humidity in the wet season originates in the Amazon Basin, due to stronger easterly winds and eastward upslope flow (especially enhanced during La Niña conditions). The link between local dust sources and easterly upslope flow is not immediately obvious. Your hypothesis would require large-scale convective processes also affecting the Altiplano. Do you have indications for that? Your precipitation data show the opposite. Hurley et al. (2015, 2016) offer a different hypothesis for depleted stable isotope ratios, i.e. the amount effect is associated with South American cold air incursions, linking synoptic-scale disturbances and monsoon dynamics to tropical ice core $\delta$18O.

Have you considered that as potential explanation for dust mobilization/uplift? How was the attribution to wet and dry season or even DJF, JJA for the ice core values conducted? This is critical and needs to be explained.

Specific comments

Lines 44-45: while moist air advection from the east is suppressed.

L 95: explain ice layer formation

L 110: Give some more details about the standard protocol: how is settling of giant particles prevented, stirring, the mean standard deviation of what?

L 137: LOD: concentration in dust or ice sample?

L 185: Calcium carbonate is also soluble in water (solubility 13 mg/l), and most likely therefore not detected in mineralogical analyses.

L 199: Massive dust deposition – I think this is exaggerated. Have you compared dust concentrations at Illimani with that in other high-alpine ice cores? Are the dust layers visible in the core?

L 500: delete percentage

Fig. 2: Show PC1 separately, in the current figure it is difficult to distinguish $\delta$D and GPPms.

Table S2: Give also ice core concentrations for comparison with other publications.

References: Hurley, J. V., et al. (2016), Forward modeling of $\delta$18O in Andean ice cores, Geophysical Research Letters, 43(15), 8178-8188.

Hurley, J. V., et al. (2015), Cold air incursions, $\delta$18O variability, and monsoon dynamics associated with snow days at Quelccaya Ice Cap, Peru, Journal of Geophysical Research, Atmospheres, 120, 7467–7487.

---

## Author Comment (AC1) · 22 Jun 2020

**Response to Anonymous Reviewer #1 to manuscript TC-2020-55-RC1**

**Title:** Giant dust particles at Nevado Illimani: a proxy of summertime deep convection over the Bolivian Altiplano

**Authors:** Filipe G. L. Lindau, Jefferson C. Simões, Barbara Delmonte, Patrick Ginot, Giovanni Baccolo, Chiara I. Paleari, Elena Di Stefano, Elena Korotkikh, Douglas S. Introne, Valter Maggi, Eduardo Garzanti, Sergio Andò

We thank the Anonymous Referee #1 for his constructive comments and suggestions. All line numbers correspond to the discussion paper and all added texts to the discussion paper are marked blue.

Specific Comment #1: Lines 44-45: while moist air advection from the east is suppressed.

[Answer]: We fixed this sentence.

Specific Comment #2: L 95: explain ice layer formation

**[Answer]: We changed the text to:**

**L 95:** Ice layers (less than 5 cm thick) occurred frequently along the firn core (Fig. S1), generally in closely spaced groups of 2 or 3 individual layers. In addition, less than 1 cm thick ice layers (or possibly wind crusts, hardly distinguishable from the former at greater depths) commonly occurred along the core. These features indicate few events of meltwater percolation, and ensure the proxies recorded in the firn core are well-preserved.

**Specific Comment #3:** L 110: Give some more details about the standard protocol: how is settling of giant particles prevented, stirring, the mean standard deviation of what?

**General Comment #1:** The correlation between giant particles and  $\delta D$ , considered as proxy for convection is based on the relative mass proportion of giant particles. Considering the fact that determining giant particles in a liquid sample is tricky, because they tend to settle, you need to substantiate the robustness of your results better. Specify if the given mean standard deviation between the two measurements with the coulter counter (lower than 3%) applies also to the giant particles alone.

**[Answer]:** Lower uncertainties were obtained for number distribuitions. Therefore, we converted (along all the text) the proportions of giant particles in terms of mass (GPPms) to terms of number (GPPnb). In order to explain that, we changed the Section 2.2 to:

**L 105:** Samples were melted at room temperature, and a ~10 mL aliquot from each was transferred to an Accuvette Beckman Coulter vial, previously washed with Millipore Q-POD® Element ultra-pure water (in an ISO 5 Class laminar flow bench located inside an ISO 6 Class clean laboratory). Each sample was treated following standardized protocols (Delmonte et al., 2002). A Beckman Coulter Multisizer 4 equipped with a 100  $\mu$ m orifice was used to measure dust concentration and grain size (400 size channels within the 2–60  $\mu$ m interval

of spherical equivalent diameter). Samples were continuously stirred until the moment of the analysis, as the larger particles tend to settle rapidly. Systematic analysis of ultra-pure water blanks allows estimating a mean signal to noise ratio around 97. Each sample was measured twice, consuming 0.5 mL per measurement. The mean relative standard deviation (RSD) between these two measurements considering both the number and the mass of particles was 7% and 29%, respectively.

The higher deviation for the mass in comparison to the total number of particles was expected due to the presence of heavy giant particles having diameters >20  $\mu$ m (coarse silt), for which small differences in size estimation lead to higher uncertainties. Indeed, when considering only the giant particles the mean RSDs were 55% and 63% for the number and mass distributions, respectively. Thus, the proportion (%) of giant particles (GPPnb) as well as total particle concentration, were calculated from the number size distribution. Approximately 14% of the samples showed very large uncertainties (RSD >100%) for GPPnb and were discarded. The mean RSD for GPPnb was 45%.

**Specific Comment #4: L 137: LOD: concentration in dust or ice sample?**

**[Answer]: We changed to:**

**L 136:** Errors for the elemental concentrations in our samples ranged from 3% for La to 17% for Cs, and the detection limits ranged from 0.1  $\mu$ g per gram of dust for Sm to 7  $\mu$ g g-1 for Ce (Table S2).

**General Comment #2: Show examples of the size distributions for the wet and dry season.**

[Answer]: We added a new Figure to the Supplement (Fig. S2):

Figure S2: The number size distribution of a typical sample from the (a) dry and (b) wet season. Red areas highlight the giant particles (between 20 and 60  $\mu$ m).

**Specific Comment #5:** Fig. 2: Show PC1 separately, in the current figure it is difficult to distinguish  $\delta D$  and GPPms.

**General Comment #3:** Show in addition the record of mass or number of giant particles for every sample. From the presented data it is unclear if the number or mass of giant particles also has a seasonal or any other variability or not.

Since total particle mass concentrations are low during the wet season (when you observe the correlation), the relative mass proportion is the ratio of two small numbers, probably having a large uncertainty. Add uncertainty bars.

[Answer]: We changed Fig. 2 by adding the number of giant particles together with the total number of particles. This is described in a new paragraph (L 181). In addition, we transferred PC1 to a new Figure (Fig. 3).

Figure 2: Dating of the Nevado Illimani firn core by annual layer counting (ALC) based on different proxies discussed in the text: (a)  $\delta D$ , (b) ionic Calcium, and (c) total and giant dust particles concentrations (light and dark gray, respectively, both are in logarithmic scale). Gray shaded vertical bands correspond to the dry season for each calendar year. All data are reported as 3-point running average of data re-sampled at 0.05 m w.eq.

**L 181:** By considering just the giant particles we also observed a seasonal pattern, with median concentrations of 15 part. mL-1 during the wet season and 30 part. mL-1 during the dry season. The well-defined oscillatory pattern of dust concentration variability reflects the extreme seasonality of precipitation over both local and regional dust sources, and the succession of dry and wet conditions. Therefore, each sample was classified as belonging to the wet or to the dry season according to dust concentration. Sublimation has a limited influence to this seasonality (Ginot et al., 2002).

---

## Referee Comment (RC2) · Anonymous Referee #2 · 5 Oct 2020

Review of Lindau et al 2020 "Giant dust particles at Nevada Illimani: a proxy. . ..." 02 October 2020 The Cryosphere

The authors present a record of ice core geochemistry and particle size data from Nevada Illimani, and ascribe the presence of very large particles to atmospheric convection processes. The correlation between the large particle data and the stable isotope is indeed surprising, and of course I respect the field and laboratory work involved in developing such records. But to me the correlation does not prove that the mechanism previously developed to explain the isotope variability at the site (convection) need necessarily apply to the particle data. I do not feel that any of the explanations

provided for the isotope or particle interpretations stand on their own, and the correlation itself is not sufficient for me to accept the particle-convective activity link. I outline my specific concerns below.

Line 70: The authors state that this is the first time giant particles have been addressed in Andean ice. Is this the first time they have been observed, or simply the first time they have been interpreted? If it's the first time they have been observed, why is that? Is there something special about the analysis here, or site, or time period, that would be unique? I cannot recall ever seeing such large particles presented and interpreted at any high elevation or polar ice core site, so some kind of brief literature review would be helpful. Have such particles ever been observed and explained at high elevation observation stations? Any evidence that would corroborate the existing and transport mechanisms would help orient the reader to what might be happening here.

Line 150: The authors assert that any particles larger than 5 microns are of obvious local origin. I have a hard time understanding what is local and what is distant in this geographic context. If local means very close (I.e., within the glacier basin) to the ice core site, then how does a convective mechanism make sense? Local wind stress seems a much likelier scenario. A better location map showing dust sources and some kind of proposed transport pathway would be helpful. And because the focus of the paper is on the giant particles, why not collect mineralogy data on them? It seems logical to compare/contrast the fine and giant particles to establish some basis for local/remote origin.

Lines 200-215: The entire logic of the paper relies on the explanation of the stable isotope signal in terms of convective activity. Yet, the paragraph begins by stating the Andean isotope signal has divergent explanations. Has the convective explanation been explicitly tested at Nevado Illimani? And do the amount effect and convective explanations occur in unison, or are they mutually exclusive? Sentences on modeling and satellite observations are interesting, but it is not clear to me that this has been tested and verified at Illimani with in situ data. Merely stating that you assume convection is

the dominant control (lines 213-214) is, to me, not sufficient in this case.

Line 216: I would like to see a better statistical treatment of the data throughout the paper. The authors note a "close correspondence" of stable isotopes and giant particles in Fig. 2a. To my eye, the r value must be at least -0.8, which is remarkable in ice core data that normally have a fairly low signal/noise ratio. I don't understand what the PCA value adds.

Lines 223-225: I don't understand the relationship among GPP, Ca, and total dust concentration. The correlation between total dust, Ca, and GPP appears poor (statistics would help confirm this), particularly in the dry season. The dry season values are quite variable in dust and Ca, but quite consistent in GPP. What is responsible, and how does it bear on local vs. regional sources?

Lin 279-280: Again, correlation does not mean that the convection hypothesis is confirmed for either isotopes or giant particles.

Line 281: The poor correlations with dry season values may be driven by the single outlier (upper left portion of Fig. 5). I would like to see the r values in Table 2 reported with and without that outlier. I suspect the story may change significantly.

Lines 290-320: I am struggling to understand the link between convective precipitation and the meteorological data. I would think that convective activity is by nature episodic, so how does monthly precipitation data accurately capture this? And how does one then link convective activity to the entrainment of dust? If it's raining hard from convective activity, how does one get giant particles into the upper atmosphere? And then, without some kind of mechanistic link, I think it is a stretch to conclude that GPP=convective activity=La Nina years.
* * *

---

## Author Comment (AC2) · 5 Nov 2020

**Response to Anonymous Reviewer #2 to manuscript TC-2020-55**

**Title:** Giant dust particles at Nevado Illimani: a proxy of summertime deep convection over the Bolivian Altiplano

**Authors:** Filipe G. L. Lindau, Jefferson C. Simões, Barbara Delmonte, Patrick Ginot, Giovanni Baccolo, Chiara I. Paleari, Elena Di Stefano, Elena Korotkikh, Douglas S. Introne, Valter Maggi, Eduardo Garzanti, Sergio Andò

We thank the Anonymous Referee #2 for his comments and suggestions. All line numbers correspond to the discussion paper and all added texts to the discussion paper are marked blue.

**Specific Comment #1: L 70:** The authors state that this is the first time giant particles have been addressed in Andean ice. Is this the first time they have been observed, or simply the first time they have been interpreted? If it's the first time they have been observed, why is that? Is there something special about the analysis here, or site, or time period, that would be unique? I cannot recall ever seeing such large particles presented and interpreted at any high elevation or polar ice core site, so some kind of brief literature review would be helpful. Have such particles ever been observed and explained at high elevation observation stations? Any evidence that would corroborate the existing and transport mechanisms would help orient the reader to what might be happening here.

**[Answer]:** *We changed the text in order to fill this gap of information:*

**L 70:** An interesting result concerns the presence of giant dust particles (presenting a diameter larger than 20 µm), whose variability is correlated with the stable isotope record. Very large mineral dust particles were generally neglected in climate studies and underrepresented or non-represented in global climate models, because of their generally local origin with respect to the sampling site, and their relatively low number concentration (Albani et al., 2014; Adebiyi and Kok, 2020). The recent observation of such large dust grains even at great distance from the source puts into question the physical models used to estimate settling velocities, and suggests some additional mechanisms such as strong turbulence and upper-level outflow are needed to keep these dust particles aloft (van der Does et al., 2018). As a consequence, there is now a growing interest in such relatively less abundant but volumetrically important dust grains, which can play an important role in biogeochemical cycles, in cloud microphysics, in the ocean carbon cycle and atmospheric radiation budget (van der Does et al., 2018; Ryder et al., 2019). A few studies have also considered large mineral particles in snow and ice, obtaining interesting results, in particular related to the relationships existing between coarse particles and the atmospheric patterns responsible for their deflation, transport and deposition (Kutuzov et al., 2016; Wu et al., 2009, 2010; Simonsen et al., 2019).

Our data show that the concentration of giant dust particles into firn is correlated with regional meteorological observations, and in particular with atmospheric deep convection over the Bolivian Altiplano during summer. This study shows for the first time that the presence of giant dust particles in Andean firn and ice is strictly

controlled by climatic processes. We found clear evidence that the convective activity over the Altiplano, reconstructed through the analysis of giant particles, is enhanced during summer periods and in particular during La Niña years, in agreement with observations concerning atmospheric circulation anomalies in the area (Vuille, 1999).

**Specific Comment #2: L 150:** The authors assert that any particles larger than 5 microns are of obvious local origin. I have a hard time understanding what is local and what is distant in this geographic context. If local means very close (i.e., within the glacier basin) to the ice core site, then how does a convective mechanism make sense? Local wind stress seems a much likelier scenario. A better location map showing dust sources and some kind of proposed transport pathway would be helpful. And because the focus of the paper is on the giant particles, why not collect mineralogy data on them? It seems logical to compare/contrast the fine and giant particles to establish some basis for local/remote origin.

**[Answer]:** *We added a definition of local and regional sources in Sect. 1:*

**L 55:** Previous ice core studies from the Central Andes (Correia et al., 2003; Knüsel et al., 2005; Osmont et al., 2019) reveal that the aerosol content of ice is dominated by local (i.e. glacier basins from Nevado Illimani) and regional (the Altiplano's area) mineral dust during the winter, when black carbon from biomass burning in the Amazon basin is also present.

**[Answer]:** *Then, we added the potential dust sources, discussed in Sect. 3.2 of the discussion paper, to Fig. 1:*

[Figure]

**Figure 1: Location of the Nevado Illimani, the Zongo Valley and the Quelccaya ice cap.** **The numbers indicate the location of the meteorological stations used for comparison with our results: 1 – El Alto; 2 – Calacoto; 3 – Patacamaya; 4 – Oruro and 5 – Potosi. The red area indicates the Altiplano-Puna Volcanic Complex (Lindsay et al., 2001). The gray squares and gray triangles denote to potential dust source areas in the salars of the Altiplano and in the Puna, respectively (Gaiero et al., 2013). The land basemap was derived from satellite data (Natural Earth I with Shaded Relief from http://www.naturalearthdata.com).**

**[Answer]:** *Finally, we fixed the text in Sect. 2.4:*

**L 150:** We used single-grain Raman spectroscopy to identify mineralogy of dust particles having a diameter smaller than 5 µm. This because this kind of analysis was carried out for provenance purposes, thus considering particles expected to travel over longer distances.

**Specific Comment #3: L 200–215:** The entire logic of the paper relies on the explanation of the stable isotope signal in terms of convective activity. Yet, the paragraph begins by stating the Andean isotope signal has divergent explanations. Has the convective explanation been explicitly tested at Nevado Illimani? And do the amount effect and convective explanations occur in unison, or are they mutually exclusive? Sentences on modeling and satellite observations are interesting, but it is not clear to me that this has been tested and verified at Illimani with in situ data. Merely stating that you assume convection is the dominant control (lines 213-214) is, to me, not sufficient in this case.

**[Answer]:** *The "amount effect" and the convective hypothesis occur in unison and therefore are not mutually exclusive. This is known from the literature, especially in the Illimani area. Indeed, climatic controls on the isotope signal in precipitation were deeply studied nearby Nevado Illimani, at the Zongo Valley (about 60 km northwest Nevado Illimani). Over Zongo, deep atmospheric convection was found to be an important control on δD during the rainy season (Vimeux et al., 2011), and represents the primary factor controlling δD, in addition to the cumulative rainfall. This gives further support to our conclusions. We have modified the text as follows:*

**L 200–215:** Seasonal variations of the water stable isotopes in snow precipitated over the Andes are also useful for dating. However, the Andean isotopic signal led to divergent interpretations (Vimeux et al., 2009). Whereas in polar ice cores the water isotopic signature is chiefly related to temperature (Uemura et al., 2012), the isotopic composition of tropical precipitation can be affected by a larger number of factors (Hoffmann et al., 2003). It is well known that the so called "amount effect" leads to an anti-correlation between the amount of precipitation and the proportion of heavier isotopes in the precipitation. This effect is in turn related to an ensemble of physical and microphysical processes producing a robust signal on the isotopic composition of precipitation (Dansgaard, 1964; Risi et al., 2008; Vuille et al., 2003). In this context, deep atmospheric convection also plays a role on stable isotope composition (Vimeux et al., 2005). Along the Zongo Valley (Fig. 1, located nearby Nevado Illimani), in particular during the summer season, the cumulative rainfall along air mass trajectory is a second order parameter in the control of isotopic depletion, being it primarily modulated by regional convective activity (Vimeux et al., 2011). In agreement, modeling studies (e.g. Bony et al., 2008; Risi et al., 2008) reveal that the stronger the convective activity during a particular event, the higher the total amount of precipitation and thus the more depleted the isotopic composition of precipitation. In addition, satellite data (Samuels-Crow et al., 2014) reveal that during the summer season the isotopic composition of water vapor strongly depends on convective activity. These observations lead the authors to conclude that the isotopic composition of snow from the tropical Andes mainly reflects tropical convection.

**Specific Comment #4: L 216:** I would like to see a better statistical treatment of the data throughout the paper. The authors note a "close correspondence" of stable isotopes and giant particles in Fig. 2a. To my eye, the r value must be at least -0.8, which is remarkable in ice core data that normally have a fairly low signal/noise ratio. I don't understand what the PCA value adds.

**[Answer]:** *We removed the PCA from the text. Then, we changed the Fig.3 that was presented in the Response to Anonymous Reviewer #1:*

**L 216:** Interestingly, we note a close correspondence between the variability of stable isotopes and the proportion of giant particles in firn (Fig. 3): oscillations of the stable isotope record (δD) closely follow the percentage of giant dust particles (GPPnb). During the dry season, giant particles are proportionally less abundant (average GPPnb 0.5%) whereas the isotopic composition of snow is less negative (average -113‰ for δD; -15‰ for δ¹⁸O). Conversely, during the wet season when giant dust particles are at their annual maximum (average GPPnb 1%) the isotopic composition of snow is more depleted (-141‰ for δD, -18‰ for δ¹⁸O), reaching its minimum.

[Figure]

**Figure 3: Relationship over the 18-years record between the percentage of giant particles with respect to the total dust particles number (GPPnb, reverse scale) and the δD. Uncertainties for each GPPnb value (expressed by the red shaded area) are relative to the standard deviation between Coulter Counter measurements. All data are reported as 3-point running average of the data previously re-sampled at 0.05 m w.eq.**

**[Answer]:** *Although ice core data normally have a fairly low signal/noise ratio, we consider that our correlation was weaker than -0.8 due to giant particles measurement uncertainties. These uncertainties were explained in the Response to Anonymous Reviewer #1, leading to the addition of the following sentences in Sect 2.2:*

**Section 2.2:** The higher deviation for the mass in comparison to the total number of particles was expected due to the presence of heavy giant particles having diameters >20 µm (coarse silt), for which small differences in size estimation lead to higher uncertainties. Indeed, when considering only the giant particles the mean RSDs were 55% and 63% for the number and mass distributions, respectively. Thus, the proportion (%) of giant particles (GPPnb) as well as total particle concentration, were calculated from the number size distribution. Approximately 14% of the samples showed very large uncertainties (RSD >100%) for GPPnb and were discarded. The mean RSD for GPPnb was 45%.

**Specific Comment #5: L 223-225:** I don't understand the relationship among GPP, Ca, and total dust concentration. The correlation between total dust, Ca, and GPP appears poor (statistics would help confirm this), particularly in the dry season. The dry season values are quite variable in dust and Ca, but quite consistent in GPP. What is responsible, and how does it bear on local vs. regional sources?

**[Answer]:** *This part of the original discussion paper was changed in the Response to Anonymous Reviewer #1, including a new Fig. 2 (as showed below). Fig. 2 shows that giant and total particles have a similar variability during the dry season. Ca²⁺ indeed features some differences. We believe these differences seem justified considering our high temporal resolution. We expect that the composition of dust in the core is not always the same, and the relative amount of calcium ready to go into solution may be different from one sample to another according to their mineralogical composition. We have modified the text as follows:*

[Figure]

**Figure 2: Dating of the Nevado Illimani firn core by annual layer counting (ALC) based on different proxies discussed in the text: (a) δD, (b) ionic Calcium, and (c) total and giant dust particles concentrations (light and dark gray, respectively, both are in logarithmic scale). Gray shaded vertical bands correspond to the dry season for each calendar year. All data are reported as 3-point running average of data re-sampled at 0.05 m w.eq.**

**L 178:** Dust concentration varies from ~2,000 particles mL$^{-1}$ (hereafter part. mL$^{-1}$) during the wetter season, to ~10,000 part. mL$^{-1}$ during the dryer season (median values). When considering extreme values, the variation range exceeds three orders of magnitude, being the lowest concentration during the wet season 150 part. mL$^{-1}$ and the highest one during the dry season 140,000 part. mL$^{-1}$. By considering just the giant particles we also

observed a seasonal pattern, but with smaller variations. Giant particles median concentration varies from 15 part. mL$^{-1}$ during the wet season and 30 part. mL$^{-1}$ during the dry season. This suggests that giant particles deposition at the Nevado Illimani is less reduced than finer particles deposition during the wet season.

Our results are in agreement with average dust concentrations from Quelccaya ice cap during the 20[th] century, ~10,000 part. mL$^{-1}$ and ~25,000 part. mL$^{-1}$ for the size ranges of 1.6–16 µm and 0.6–20 µm, respectively (Thompson et al., 1986, 2013). The well-defined oscillatory pattern of dust concentration variability reflects the extreme seasonality of precipitation over both local and regional dust sources, and the succession of dry and wet conditions. Therefore, each sample was classified as belonging to the wet or to the dry season according to dust concentration. Sublimation has a limited influence on seasonality (Ginot et al., 2002).

The seasonality of dust concentration is in accordance with the $Ca^{2+}$ record and also with literature studies (Knüsel et al., 2005). However, both records show differences, in particular during the dry season when they are not significantly correlated at the 95% level. Considering our high temporal sampling resolution, this might be associated with slight changes in dust mineralogy, possibly affecting the amount of calcium to be solubilized. Ionic calcium can be primarily associated to calcium sulphate ($CaSO_4$) and/or calcium carbonate ($CaCO_3$) (Kutuzov et al., 2019). Because scarcity of calcium carbonates was revealed by mineralogical analyses (Fig. 4, see below), we argue that most of the ionic calcium observed in firn samples is present as a soluble species, probably $CaSO_4$, and not detectable through Raman spectroscopy on single insoluble particles. However, we consider the possibility of calcium carbonate depletion due to scavenging during dust transport and/or dissolution during the melting of the samples, as discussed by Wu et al. (2016) based on ice core samples from the Tibetan Plateau. In addition, we cannot exclude that Ca-bearing aerosols might have been initially a mixture of pure gypsum and calcium carbonates that successively reacted with atmospheric $H_2SO_4$, both in the atmosphere or within the snow-pack as the result of post-depositional processes (Röthlisberger et al., 2000; Iizuka et al., 2008).

**Specific Comment #5: L 279–280:** Again, correlation does not mean that the convection hypothesis is confirmed for either isotopes or giant particles.

**[Answer]:** *We tried to improve our discussion also considering more evidences from literature. We have modified the text as follows:*

**L 281:** Seasonal mean values (Fig. 6) show the significant correlation between these two proxies. Data located in the bottom right corner of Fig. 6 refer to dust deposited onto the Nevado Illimani glacier during the wet season, when more intense convection leads to higher percentage of giant particles and more depleted isotopic composition of precipitation. Points located in the upper left corner, conversely, are associated to winter dry periods when convection and precipitation are reduced and the relative proportion of giant particles is lower. Convective activity is known to significantly affect the isotopic composition of tropical precipitation. Intense regional convection leads to more isotopically-depleted precipitation (e.g. Risi et al., 2008). A proposed mechanism is that convective downdraft promotes the subsidence of higher level water vapor, causing isotopic depletion of low level vapor crossing the eastern Andes (Vimeux et al., 2011). Convective downdrafts, in turn, are often associated with density currents, offering an efficient mechanism for dust lifting (Flamant et al., 2007). Indeed, the leading edge of the density current is characterized by strong turbulent winds that can mobilize dust and mix it through a deep layer (Knippertz et al., 2007). In accordance, events of giant dust particles suspension and transport, as detected by aircraft measurements in north Africa, were related to the occurrence of convective systems (Ryder et al., 2013). Therefore, giant particles on the Nevado Illimani glacier can be reasonably used as

proxy for deep summer convective precipitation. Given their size and geochemical/mineralogical fingerprint, we confidently associate them to local/regional convective activity near the Bolivian Altiplano.

**Specific Comment #6: L 281:** The poor correlations with dry season values may be driven by the single outlier (upper left portion of Fig. 5). I would like to see the r values in Table 2 reported with and without that outlier. I suspect the story may change significantly.

**[Answer]:** *There is indeed a significant correlation during the dry season. This was adjusted in the Response to Anonymous Reviewer #1, where both Fig. 5 (hereafter Fig. 6) and Table 2 were changed:*

[Figure]

Figure 6: Seasonal mean GPPnb and δD for the dry seasons (orange circle) and the wet season (blue circle). Error bars (horizontal bands) for GPPnb are based on the mean relative standard deviation for the samples integrating each season. Light gray dots on background are raw data. The La Niña events discussed in the text (1999-00, 2000-01, 2007-08, and 2010-11) are reported.

Table 2: Seasonal correlations between Giant Particles Percentage (GPPnb) and δD, rainfall observations and outgoing longwave radiation (OLR). The period between December and February was defined as the wet season, and the period between June and August as the dry season. Significant correlations at 95% level are shown in bold.

| GPPnb | δD | El Alto | Calacoto | Patacamaya | Oruro | Potosi | OLR |
|-------|-------|---------|----------|------------|-------|--------|-------|
| Wet | **-0.71** | 0.47 | 0.39 | **0.76** | 0.25 | 0.41 | **-0.69** |
| Dry | **-0.70** | 0.04 | -0.21 | -0.09 | -0.08 | -0.23 | 0.07 |

**[Answer]:** *Then we explained the dry season relationships in the following paragraph:*

**L 306:** We conclude that the more intense is summer convection, the higher is the relative number of giant dust particles suspended in the atmosphere and the more depleted is the δD. Winter δD, conversely, is more influenced by the local amount effect, as re-evaporation plays a greater role in such a dry environment (Vimeux

et al., 2011). Although we observed a significant correlation between dry season GPPnb and δD, the rare winter convection seems also to have a low influence in GPPnb variability, as indicated by its lack of significant correlations with both JJA precipitation and OLR (Table 2).

**Specific Comment #7: L 290−320:** I am struggling to understand the link between convective precipitation and the meteorological data. I would think that convective activity is by nature episodic, so how does monthly precipitation data accurately capture this? And how does one then link convective activity to the entrainment of dust? If it's raining hard from convective activity, how does one get giant particles into the upper atmosphere? And then, without some kind of mechanistic link, I think it is a stretch to conclude that GPP=convective activity=La Nina years.

**[Answer]:** *First, we changed the text in order to explain the relationship between convective activity and precipitation:*

**L 290:** In order to test the hypothesis of a relationship between giant particles and convective precipitation, we analyzed monthly precipitation and wind speed records from five meteorological stations located in the central Andes (Fig. 1). Data was provided by SENAMHI, Bolivia (www.senamhi.gob.bo/sismet), whereas monthly outgoing longwave radiation (OLR) data on a 2.5° x 2.5° grid box (Liebmann and Smith, 1996) was obtained from NOAA/OAR/ESRL PSD, Boulder, Colorado, USA (https://www.esrl.noaa.gov/psd/). Low OLR values correspond to cold and high clouds which denote enhanced convection. OLR data centered at 17.5°S, 70°W was used as an index of the convective precipitation over the Altiplano, as it presents strong negative correlations with regional rainfall observations (Garreaud and Aceituno, 2001). In addition, both OLR and precipitation data provided similar results when linking δD and regional convection at the Zongo Valley (Vimeux et al., 2011). In fact, it is estimated that deep convection provides 65% of the precipitation over this region, as the orographic lifting of moisture from the Amazon basin through Andes trigger condensation, latent heat release, and strong convective updrafts during the summer (Insel et al., 2010).

Meteorological and OLR datasets were re-sampled into DJF (December to February) and JJA (June to August) time series and compared with our seasonally resolved GPPnb series. For each wet and dry season, defined by dust concentration variability (Sect. 3.1), a mean GPPnb was obtained.

**[Answer]:** *Then, we have modified the paragraph explaining the possible relations with La Niña as follows:*

Fig. 6 shows that over the 18-years period analyzed in this work, the summer seasons showing the most intense levels of convection (considering both GPPnb and δD) were 2000−2001 and 2010−2011. Both correspond to La Niña periods, as indicated by their DJF Oceanic Niño Index (ONI) of -0.7 and -1.4, respectively. It is well known that the El Niño-Southern Oscillation phenomenon has a significant impact on climate over the Altiplano, especially during the summer season. In particular, meteorological data show that La Niña conditions intensify the meridional pressure gradient on the northern side of the Bolivian High, leading to stronger high troposphere easterly winds, increased eastward upslope flow and enhanced moisture transport (Garreaud, 1999; Vuille, 1999). However, the strong DJF La Niña events of 1999−2000 (ONI = -1.7), and 2007−2008 (-1.6) do not show higher GPPnb or more depleted δD compared to other values of the wet season (Fig. 6). We believe this was due to competing mechanisms controlling moisture transport from the Amazon basin to the Altiplano. In addition to the role played by the upper troposphere easterly winds, the meridional circulation between the tropical North Atlantic Ocean and the western tropical South America also influences the DJF precipitation over the Central Andes, especially in the last two decades (Segura et al., 2020). Evidence based on reanalysis data indicates that

when this meridional circulation is enhanced, the atmospheric stability between the mid and the upper troposphere over the Altiplano is reduced, resulting in increased moisture transport from the Amazon basin (Segura et al., 2020). Thus, we propose a new approach for future studies in tropical Andean glaciers based on giant particles and stable isotopes of snow. This can be used as a complement to a number of other climate proxies and modeling experiments, providing insights into past atmospheric circulation over tropical South America.

**General Comment #1:** I do not feel that any of the explanations provided for the isotope or particle interpretations stand on their own, and the correlation itself is not sufficient for me to accept the particle-convective activity link.

**[Answer]:** *Our particle and isotope interpretations are corroborated by regional meteorological data and also by previous studies carried out near the Illimani. Also, both particle-convection and isotopes-convection relationships are in agreement with studies specifically dealing with these links. Our study is a first exploratory work interpreting the giant dust particles content in a tropical Andean firn core, therefore we believe stronger correlations might be obtained by reducing uncertainties (which are extensively discussed in our manuscript) in giant particles measurements, but this is something that we will explore in the future and goes beyond the aim of this work.*

**Added references**

Adebiyi, A. A. and Kok, J. F.: Climate models miss most of the coarse dust in the atmosphere, Sci. Adv., 6(15), 1–10, doi:10.1126/sciadv.aaz9507, 2020.

Albani, S., Mahowald, N. M., Perry, A. T., Scanza, R. A., Zender, C. S., Heavens, N. G., Maggi, V., Kok, J. F. and Otto-Bliesner, B. L.: Improved dust representation in the Community Atmosphere Model, J. Adv. Model. Earth Syst., 6, 541–570, doi:10.1002/2013MS000279, 2014.

van der Does, M., Knippertz, P., Zschenderlein, P., Giles Harrison, R. and Stuut, J. B. W.: The mysterious long-range transport of giant mineral dust particles, Sci. Adv., 4(12), 1–9, doi:10.1126/sciadv.aau2768, 2018.

Flamant, C., Chaboureau, J.-P., Parker, D. J., Taylor, C. M., Cammas, J.-P., Bock, O., Timouk, F. and Pelon, J.: Airborne observations of the impact of a convective system on the planetary boundary layer thermodynamics and aerosol distribution in the inter-tropical discontinuity region of the West African Monsoon, Q. J. R. Meteorol. Soc., 133, 1175–1189, doi:10.1002/qj, 2007.

Iizuka, Y., Horikawa, S., Sakurai, T., Johnson, S., Dahl-Jensen, D., Steffensen, J. P. and Hondoh, T.: A relationship between ion balance and the chemical compounds of salt inclusions found in the Greenland Ice Core Project and Dome Fuji ice cores, J. Geophys. Res. Atmos., 113(7), 1–11, doi:10.1029/2007JD009018, 2008.

Insel, N., Poulsen, C. J. and Ehlers, T. A.: Influence of the Andes Mountains on South American moisture transport, convection, and precipitation, Clim. Dyn., 35(7), 1477–1492, doi:10.1007/s00382-009-0637-1, 2010.

Knippertz, P., Deutscher, C., Kandler, K., Müller, T., Schulz, O. and Schütz, L.: Dust mobilization due to density currents in the Atlas region: Observations from the Saharan Mineral Dust Experiment 2006 field campaign, J. Geophys. Res. Atmos., 112(21), 1–14, doi:10.1029/2007JD008774, 2007.

Kutuzov, S. S., Mikhalenko, V. N., Grachev, A. M., Ginot, P., Lavrentiev, I. I., Kozachek, A. V., Krupskaya, V. V., Ekaykin, A. A., Tielidze, L. G. and Toropov, P. A.: First geophysical and shallow ice core investigation of the Kazbek plateau glacier, Caucasus Mountains, Environ. Earth Sci., 75(23), doi:10.1007/s12665-016-6295-9, 2016.

Ryder, C. L., Highwood, E. J., Lai, T. M., Sodemann, H. and Marsham, J. H.: Impact of atmospheric transport on the evolution of microphysical and optical properties of Saharan dust, Geophys. Res. Lett., 40(10), 2433–2438, doi:10.1002/grl.50482, 2013.

Ryder, C. L., Highwood, E. J., Walser, A., Seibert, P., Philipp, A. and Weinzierl, B.: Coarse and giant particles are ubiquitous in Saharan dust export regions and are radiatively significant over the Sahara, Atmos. Chem. Phys., 19(24), 15353–15376, doi:10.5194/acp-19-15353-2019, 2019.

Simonsen, M. F., Baccolo, G., Blunier, T., Borunda, A., Delmonte, B., Frei, R., Goldstein, S., Grinsted, A., Kjær, H. A., Sowers, T., Svensson, A., Vinther, B., Vladimirova, D., Winckler, G., Winstrup, M. and Vallelonga, P.: East Greenland ice core dust record reveals timing of Greenland ice sheet advance and retreat, Nat. Commun., 10(1), doi:10.1038/s41467-019-12546-2, 2019.

Vimeux, F., Tremoy, G., Risi, C. and Gallaire, R.: A strong control of the South American SeeSaw on the intra-seasonal variability of the isotopic composition of precipitation in the Bolivian Andes, Earth Planet. Sci. Lett., 307(1–2), 47–58, doi:10.1016/j.epsl.2011.04.031, 2011.

Wu, G., Zhang, C., Gao, S., Yao, T., Tian, L. and Xia, D.: Element composition of dust from a shallow Dunde ice core, Northern China, Glob. Planet. Change, 67(3–4), 186–192, doi:10.1016/j.gloplacha.2009.02.003, 2009.

Wu, G., Yao, T., Xu, B., Tian, L., Zhang, C. and Zhang, X.: Dust concentration and flux in ice cores from the Tibetan Plateau over the past few decades, Tellus, Ser. B Chem. Phys. Meteorol., 62(3), 197–206, doi:10.1111/j.1600-0889.2010.00457.x, 2010.

---

## Author Response (AR1)

**Response to Anonymous Reviewers #1 and #2 to manuscript TC-2020-55**

**Title:** Giant dust particles at Nevado Illimani: a proxy of summertime deep convection over the Bolivian Altiplano

**Authors:** Filipe G. L. Lindau, Jefferson C. Simões, Barbara Delmonte, Patrick Ginot, Giovanni Baccolo, Chiara I. Paleari, Elena Di Stefano, Elena Korotkikh, Douglas S. Introne, Valter Maggi, Eduardo Garzanti, Sergio Andò

This document shows a point-by-point reply to comments from both reviewers (RC1 and RC2). It is followed by a marked-up version of the manuscript showing the changes we have made. All line numbers correspond to the marked-up manuscript.

**RC1, Specific Comment #1: Line 40:** while moist air advection from the east is suppressed.

[Answer]: We fixed this sentence.

**RC2, Specific Comment #1: L 79–80:** The authors state that this is the first time giant particles have been addressed in Andean ice. Is this the first time they have been observed, or simply the first time they have been interpreted? If it's the first time they have been observed, why is that? Is there something special about the analysis here, or site, or time period, that would be unique? I cannot recall ever seeing such large particles presented and interpreted at any high elevation or polar ice core site, so some kind of brief literature review would be helpful. Have such particles ever been observed and explained at high elevation observation stations? Any evidence that would corroborate the existing and transport mechanisms would help orient the reader to what might be happening here.

[Answer]: We changed the text in order to fill this gap of information.

**RC1, Specific Comment #2: L 105: Explain ice layer formation**

[Answer]: We added an explanation about ice layer formation to the text.

**RC1, Specific Comment #3: L 120:** Give some more details about the standard protocol: how is settling of giant particles prevented, stirring, the mean standard deviation of what?

**RC1, General Comment #1:** The correlation between giant particles and  $\delta D$ , considered as proxy for convection is based on the relative mass proportion of giant particles. Considering the fact that determining giant particles in a liquid sample is tricky, because they tend to settle, you need to substantiate the robustness of your results better. Specify if the given mean standard deviation between the two measurements with the coulter counter (lower than 3%) applies also to the giant particles alone.

**[Answer]:** We added details about the standard protocol to the text. Then, we added a discussion about the deviations between the measurements for the giant particles. Number distributions showed a lower uncertainty than mass distributions. Therefore, we converted (along all the text) the proportions of giant particles in terms of mass (GPPms) to terms of number (GPPnb).

RC1, Specific Comment #4: L 155: LOD: concentration in dust or ice sample?

[Answer]: We changed the text in order to fill this gap of information.

**RC1, General Comment #2:** Show examples of the size distributions for the wet and dry season.

[Answer]: We added a new Figure to the Supplement (Fig. S2) showing these examples.

**RC2, Specific Comment #2: L 170–171:** The authors assert that any particles larger than 5 microns are of obvious local origin. I have a hard time understanding what is local and what is distant in this geographic context. If local means very close (i.e., within the glacier basin) to the ice core site, then how does a convective mechanism make sense? Local wind stress seems a much likelier scenario. A better location map showing dust sources and some kind of proposed transport pathway would be helpful. And because the focus of the paper is on the giant particles, why not collect mineralogy data on them? It seems logical to compare/contrast the fine and giant particles to establish some basis for local/remote origin.

**[Answer]:** We added a definition of local and regional sources in L 52–53. Then, we added the location of the potential dust sources, discussed in Sect. 3.2, to Fig. 1. Finally, we fixed the text in L 170–171.

**RC1, Specific Comment #5:** Fig. 2: Show PC1 separately, in the current figure it is difficult to distinguish δD and GPPms.

**RC2**, **Specific Comment #3:** I don't understand what the PCA value adds.

[Answer]: We removed the PCA from the text.

**RC1, General Comment #3:** Show in addition the record of mass or number of giant particles for every sample. From the presented data it is unclear if the number or mass of giant particles also has a seasonal or any other variability or not.

Since total particle mass concentrations are low during the wet season (when you observe the correlation), the relative mass proportion is the ratio of two small numbers, probably having a large uncertainty. Add uncertainty bars.

**[Answer]:** In Fig. 2 we plotted the number of giant particles together with the total number of particles. We have also included a discussion of the variability of its concentration in L 206–207. The proportion of giant dust particles (GPPnb) was plotted in a new Fig. 3.

**RC1, Specific Comment #6:** L **216–217**: Calcium carbonate is also soluble in water (solubility 13 mg/l), and most likely therefore not detected in mineralogical analyses.

**RC2, Specific Comment #4: L 211–222:** I don't understand the relationship among GPP, Ca, and total dust concentration. The correlation between total dust, Ca, and GPP appears poor (statistics would help confirm this), particularly in the dry season. The dry season values are quite variable in dust and Ca, but quite consistent in GPP. What is responsible, and how does it bear on local vs. regional sources?

**[Answer]:** Fig. 2 shows that giant and total particles have a similar variability during the dry season.  $Ca^{2+}$  indeed features some differences. We believe these differences seem justified considering our high temporal resolution. We expect that the composition of dust in the core is not always the same, and the relative amount of calcium ready to go into solution may be different from one sample to another according to their mineralogical composition.

**RC1, Specific Comment #7: L 232:** Massive dust deposition – I think this is exaggerated. Have you compared dust concentrations at Illimani with that in other high-alpine ice cores? Are the dust layers visible in the core?

[Answer]: We changed the word massive. Then, we compared our dust concentration record with that of another Andean ice core (Quelccaya).

**RC2, Specific Comment #5: L 234–251:** The entire logic of the paper relies on the explanation of the stable isotope signal in terms of convective activity. Yet, the paragraph begins by stating the Andean isotope signal has divergent explanations. Has the convective explanation been explicitly tested at Nevado Illimani? And do the amount effect and convective explanations occur in unison, or are they mutually exclusive? Sentences on modeling and satellite observations are interesting, but it is not clear to me that this has been tested and verified at Illimani with in situ data. Merely stating that you assume convection is the dominant control (lines 213-214) is, to me, not sufficient in this case.

**[Answer]:** The "amount effect" and the convective hypothesis occur in unison and therefore are not mutually exclusive. This is known from the literature, especially in the Illimani area. Indeed, climatic controls on the isotope signal in precipitation were deeply studied nearby Nevado Illimani, at the Zongo Valley (about 60 km northwest Nevado Illimani). Over Zongo, deep atmospheric convection was found to be an important control on  $\delta D$  during the rainy season (Vimeux et al., 2011), and represents the primary factor controlling  $\delta D$ , in addition to the cumulative rainfall. This gives further support to our conclusions. We have modified the text as follows.

**RC2, Specific Comment #6: L 259:** I would like to see a better statistical treatment of the data throughout the paper. The authors note a "close correspondence" of stable isotopes and giant particles in Fig. 2a. To my eye, the r value must be at least -0.8, which is remarkable in ice core data that normally have a fairly low signal/noise ratio.

**[Answer]:** Although ice core data normally have a fairly low signal/noise ratio, we consider that our correlation was weaker than -0.8 due to giant particles measurement uncertainties. These uncertainties were explained in L 126–131.

**RC2, Specific Comment #7: L 323–325:** Again, correlation does not mean that the convection hypothesis is confirmed for either isotopes or giant particles.

[Answer]: We tried to improve our discussion also considering more evidences from literature (L 332-339).

**RC2, Specific Comment #8: L 281:** The poor correlations with dry season values may be driven by the single outlier (upper left portion of Fig. 5). I would like to see the r values in Table 2 reported with and without that outlier. I suspect the story may change significantly.

**[Answer]:** When we started working with number distributions (GPPnb) instead of mass distributions (GPPns), we changed both Fig. 5 (hereafter Fig. 6) and Table 2. Indeed, there is a significant correlation between GPPnb and  $\delta D$  during the dry season. The dry season relationships were explained in L 381–386.

**RC1, General Comment #4:** Before being lifted up by convection, dust particles need to be mobilized from the ground, which requires strong wind (advection). Have you checked wind speeds in the dust source areas? Dust source areas are located SE of Illimani, whereas humidity in the wet season originates in the Amazon Basin, due to stronger easterly winds and eastward upslope flow (especially enhanced during La Niña conditions). The link between local dust sources and easterly upslope flow is not immediately obvious. Your hypothesis would require large-scale convective processes also affecting the Altiplano. Do you have indications for that? Your precipitation data show the opposite. Hurley et al. (2015, 2016) offer a different hypothesis for depleted stable isotope ratios, i.e. the amount effect is associated with South American cold air incursions, linking synoptic-scale disturbances and monsoon dynamics to tropical ice core  $\delta^{18}$ O. Have you considered that as potential explanation for dust mobilization/uplift? How was the attribution to wet and dry season or even DJF, JJA for the ice core values conducted? This is critical and needs to be explained.

**[Answer]:** We have added a more detailed explanation of our proposed mechanism for dust mobilization/uplift in L 358–379. A better description of the method we used to assign the wet and dry seasons to the firn core values was included in Sect. 3.1 (L 255–258).

**RC2, Specific Comment #9: L 387–408:** I am struggling to understand the link between convective precipitation and the meteorological data. I would think that convective activity is by nature episodic, so how does monthly precipitation data accurately capture this? And how does one then link convective activity to the entrainment of dust? If it's raining hard from convective activity, how does one get giant particles into the upper atmosphere? And then, without some kind of mechanistic link, I think it is a stretch to conclude that GPP=convective activity=La Nina years.

**[Answer]:** We changed the text to explain the relationship between convective activity and precipitation (L 344–350). Then, we modified the paragraph that explains the possible relations with La Niña (L 387–408).

**RC2, General Comment #1:** I do not feel that any of the explanations provided for the isotope or particle interpretations stand on their own, and the correlation itself is not sufficient for me to accept the particle-convective activity link.

[Answer]: Our particle and isotope interpretations are corroborated by regional meteorological data and also by previous studies carried out near the Illimani. Also, both particle-convection and isotopes-convection relationships are in agreement with studies specifically dealing with these links. Our study is a first exploratory work interpreting the giant dust particles content in a tropical Andean firn core, therefore we believe stronger correlations might be obtained by reducing uncertainties (which are extensively discussed in our manuscript) in giant particles measurements, but this is something that we will explore in the future and goes beyond the aim of this work.

**RC1, Specific Comment #8**: Table S2: Give also ice core concentrations for comparison with other publications.

[Answer]: We added a comparison in Table S2.

**Giant dust particles at Nevado Illimani: a proxy of summertime deep convection over the Bolivian Altiplano**

Filipe G. L. Lindau1, Jefferson C. Simões1,2, Barbara Delmonte3, Patrick Ginot4, Giovanni Baccolo3, Chiara I. Paleari3, Elena Di Stefano3, Elena Korotkikh2, Douglas S. Introne2, Valter Maggi3, Eduardo Garzanti3, and Sergio Andò3

[revised manuscript text omitted]

---

## Referee Report (RR1)

Second review of Lindau et al 2020
"Giant dust particles at Nevada Illimani: a proxy…."
21 December 2020
The Cryosphere

This is my second review of this paper, and considers the revised version the authors submitted and their responses to my (and the other reviewer's) comments from the first version.

I have a number of remaining concerns about the manuscript, and the author's responses to the original reviews.

Lines 100-104 - Ice layers in the core. The other reviewer pointed this out in the first round of reviews, and the authors added a sentence or two in response (and a very general figure in the supplementary material). I think their analysis falls far short of what is required to justify interpreting the particle record as a purely climate signal. The author's claim that "these features indicate few events of meltwater percolation" (line 102) is not backed up by any analysis. I would need to see some sort of analysis of the various core proxies vs. the ice layer record to have any confidence in that statement. One obvious question - what is the effect of melting on the particle size distribution? Is the depth variability of the giant particle concentration simply a function of surface concentration during melting?

Lines 125 (and throughout) - The use of GPPnb as a proxy. The authors present a timeseries of particle concentration (Fig. 3) and interpret the seasonal pattern of both total and giant particle deposition. In this figure, and lines 200-201, the authors clearly show that giant particle concentrations are highest in the dry season, and much lower (by at least a factor of 2) during the wet season during convective activity. I would argue that this (giant particle concentration) is the most accurate measure of giant particle deposition at the site (after, of course, the authors answer the post-depositional modification question). Yet, the authors then proceed to move to a relative measure of giant particles (giant particle percentage GPP). Unfortunately, that measure conflates two uncertain measures - both fine and giant particles. What if atmospheric processes were affecting the two differently, such that GPP is being altered primarily by fine grain processes? At the very least, I would have to see both measures (GP concentration and GPP) statistically compared vs. the other core parameters and meteorological variables. I suspect (and could of course be wrong) that the correlations with GPP could be non-existent or even absent when run with GP concentrations.

Figures - Figure 3 is not relevant to the author's argument. The overall correlation through the entire record (which the authors have still not quantified) has no bearing on the wet season convective activity. A much more useful figure would be wet season dD, wet season giant particle concentration AND wet season GPPnb vs. year. Similarly, Figure 6 needs to include giant particle concentration for dry and wet seasons vs. dD (not just GPPnb).

Dust provenance - I'm still confused as to what the dust provenance work is supposed to show. The focus here is on the giant particles, and the question that remains to me is - are these

particles simply local (in the Illimani massif) or from some farther source?  The simplest explanation seems to be that these giant particles are simply local transport and thus do not require a complicated convective activity explanation.  But that can only be proved if the dust provenance geochemistry clearly shows the giant particles are not of local origin.  The sample choice and data do not seem to be able to shed any light on this issue - wet vs. dry season geochemistry on bulk samples provides no specific information on the provenance of the giant particles.  The giant particles could be from right next the the drillsite, and deposited with a background, fine fraction dust matrix that is from a remote location.

---

## Author Response (AR2)

**Response to Anonymous Reviewers #1 and #2 to manuscript TC-2020-55**

**Title:** Giant dust particles at Nevado Illimani: a proxy of summertime deep convection over the Bolivian Altiplano

**Authors:** Filipe G. L. Lindau, Jefferson C. Simões, Barbara Delmonte, Patrick Ginot, Giovanni Baccolo, Chiara I. Paleari, Elena Di Stefano, Elena Korotkikh, Douglas S. Introne, Valter Maggi, Eduardo Garzanti, Sergio Andò

**This document shows a point-by-point reply to comments from both reviewers (RC1 and RC2). It is followed by a marked-up version of the manuscript showing the changes we have made. All line numbers correspond to the marked-up manuscript.**

**RC2, Specific Comment #1: L 101-104:** Ice layers in the core. The other reviewer pointed this out in the first round of reviews, and the authors added a sentence or two in response (and a very general figure in the supplementary material). I think their analysis falls far short of what is required to justify interpreting the particle record as a purely climate signal. The author's claim that "these features indicate few events of meltwater percolation" (line 104) is not backed up by any analysis. I would need to see some sort of analysis of the various core proxies vs. the ice layer record to have any confidence in that statement. One obvious question - what is the effect of melting on the particle size distribution? Is the depth variability of the giant particle concentration simply a function of surface concentration during melting?

**[Answer]:** *We changed the figure in the supplementary material (Fig. S1). Now it shows the depth variability of GPPnb and δD, and also the depth intervals where we observed ice/crust layers. This figure points to no clear relationship between ice/crust layers and these proxies. Therefore, we assume that meltwater percolation had little influence on our record. The text in Line 101 has been changed to improve this discussion.*

**RC1, General Comment #1:** The statistical treatment is still poor. With a mean RSD for GP of 45% also the percentage has a large uncertainty (error propagation). I wonder if the correlation of the GPPnb percentage with delta D is statistically different from the correlation between total particle number concentration and delta D. Since this is the main finding, it should be better supported.

**[Answer]:** *We improved the statistical treatment. First, we isolated the random components of our records by removing their seasonality and outliers. Then, we tested the distribution of the random components. Based on this test, we performed the Spearman correlation analysis. Finally, we determined the confidence intervals for each correlation, using a block bootstrap resampling method followed by the Fisher's transformation. This procedure allowed us to observe that the correlation between GPPnb and δD is statistically higher than the correlation between total particle number concentration and δD. These procedures and discussions were included in the new subsection 2.6 (Correlation evaluation), and in Line 282.*

**RC2, Specific Comment #2: Lines 133 (and throughout):** The use of GPPnb as a proxy. The authors present a time series of particle concentration (Fig. 3) and interpret the seasonal pattern of both total and giant particle deposition. In this figure, and lines 224-225, the authors clearly show that giant particle concentrations are highest in the dry season, and much lower (by at least a factor of 2) during the wet season during convective activity. I would argue that this (giant particle concentration) is the most accurate measure of giant particle deposition at the site (after, of course, the authors answer the post-depositional modification question). Yet, the authors then proceed to move to a relative measure of giant particles (giant particle percentage GPP). Unfortunately, that measure conflates two uncertain measures - both fine and giant particles. What if atmospheric processes were affecting the two differently, such that GPP is being altered primarily by fine grain processes? At the very least, I would have to see both measures (GP concentration and GPP) statistically compared vs. the other core parameters and meteorological variables. I suspect (and could of course be wrong) that the correlations with GPP could be non-existent or even absent when run with GP concentrations.

**RC2, Specific Comment #3:** Figure 3 is not relevant to the author's argument. The overall correlation through the entire record (which the authors have still not quantified) has no bearing on the wet season convective activity. A much more useful figure would be wet season dD, wet season giant particle concentration AND wet season GPPnb vs. year. Similarly, Figure 6 needs to include giant particle concentration for dry and wet seasons vs. dD (not just GPPnb).

*[Answer]: We have observed no significant correlation between the number concentration of giant particles (GP) and both δD and the meteorological variables. The reason for this is that we believe the effect of convection on GP is twofold. Giant particle suspension is favored by convective activity, on the other hand, GP increases during dry conditions by increased source strength and reduced accumulation. Conversely, GPPnb seems to provide the overbalance between turbulence and source strength/accumulation. We added this discussion to the text (Line 368). Then, we included an indication of the wet seasons in Fig. 3. This figure introduces the GPPnb time series and its relationship with δD.*

**RC2, Specific Comment #4:** Dust provenance - I'm still confused as to what the dust provenance work is supposed to show. The focus here is on the giant particles, and the question that remains to me is - are these particles simply local (in the Illimani massif) or from some farther source? The simplest explanation seems to be that these giant particles are simply local transport and thus do not require a complicated convective activity explanation. But that can only be proved if the dust provenance geochemistry clearly shows the giant particles are not of local origin. The sample choice and data do not seem to be able to shed any light on this issue - wet vs. dry season geochemistry on bulk samples provides no specific information on the provenance of the giant particles. The giant particles could be from right next the the drillsite, and deposited with a background, fine fraction dust matrix that is from a remote location.

*[Answer]: The dust provenance section shows that source areas are local/regional during both wet and dry seasons. Deviations in dust mineralogy and geochemistry seem to be associated with increased scavenging during wet seasons due to heavier precipitation. Although we have no specific data for the giant particles provenance, these conclusions support the influence of local/regional convective activity on GPPnb variability. We added to Line 333, an improved conclusion for the dust provenance section.*